# CPO: Condition Preference Optimization for Controllable Image Generation

**Zonglin Lyu**   **Ming Li**   **Xinxin Liu**   **Chen Chen**
Institute of Artificial Intelligence
University of Central Florida
Orlando, FL 32816
{zonglin.lyu, ming.li, xinxin.liu, chen.chen}@ucf.edu

## Abstract

To enhance controllability in text-to-image generation, ControlNet introduces image-based control signals, while ControlNet++ improves pixel-level cycle consistency between generated images and the input control signal. To avoid the prohibitive cost of back-propagating through the sampling process, ControlNet++ optimizes only low-noise timesteps (e.g., $t < 200$) using a single-step approximation, which not only ignores the contribution of high-noise timesteps but also introduces additional approximation errors. A straightforward alternative for optimizing controllability across all timesteps is Direct Preference Optimization (DPO), a fine-tuning method that increases model preference for more controllable images ($I^w$) over less controllable ones ($I^l$). However, due to uncertainty in generative models, it is difficult to ensure that win–lose image pairs differ only in controllability while keeping other factors, such as image quality, fixed. To address this, we propose performing preference learning over control conditions rather than generated images. Specifically, we construct winning and losing control signals, $\mathbf{c}^w$ and $\mathbf{c}^l$, and train the model to prefer $\mathbf{c}^w$. This method, which we term *Condition Preference Optimization* (CPO), eliminates confounding factors and yields a low-variance training objective. Our approach theoretically exhibits lower contrastive loss variance than DPO and empirically achieves superior results. Moreover, CPO requires less computation and storage for dataset curation. Extensive experiments show that CPO significantly improves controllability over the state-of-the-art ControlNet++ across multiple control types: over $10\%$ error rate reduction in segmentation, $70$–$80\%$ in human pose, and consistent $2$–$5\%$ reductions in edge and depth maps. The error rate is defined as the difference between the evaluated controllability and the oracle results. Our project is available here.

## 1 Introduction

Recent advancements in diffusion models [1–3] have achieved state-of-the-art performance in text-to-image (T2I) generation. Text prompts serve as sparse conditioning signals, typically conveying only global attributes such as background, foreground, and salient objects. However, when multiple objects are present or when fine-grained details, such as edges or 2D/3D spatial structure, are involved, text prompts struggle to capture such information precisely. These details are more effectively conveyed through dense visual conditions, such as segmentation maps [4], depth maps [5], and Canny edges [6]. To incorporate such dense control, several works [7–10] introduce additional modules that extract control signals on top of base T2I models, such as Stable Diffusion [2]. However, these methods do not define an explicit training objective to ensure precise controllability.

ControlNet++ [11] introduces a *pixel-level cycle consistency loss* between the generated image $\hat{\mathbf{x}}$ and the input control condition $\mathbf{c}$ to explicitly model controllability during training, as shown in

39th Conference on Neural Information Processing Systems (NeurIPS 2025).

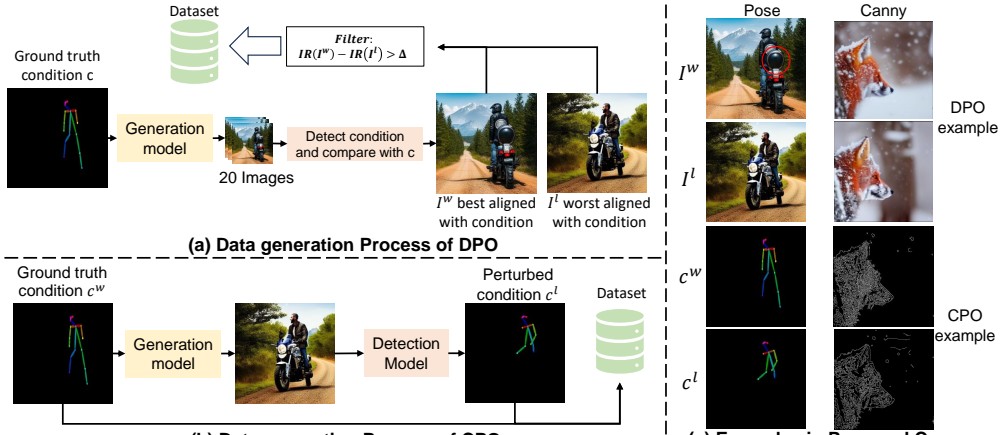

**(a) Data generation Process of DPO**

**(b) Data generation Process of CPO**

**(c) Examples in Pose and Canny**

Figure 1: (a) **Data generation Process of DPO**. We generate 20 images and find images that are the best and the worst aligned with the input condition. The ImageReward (IR) score of the winner is 0.2 higher; otherwise, the example is filtered out. (b) **Data generation process of CPO**. Our process is much simpler. We generate one image to perturb the ground truth condition since the generation model is not perfectly controllable. (c) **Examples on Pose and Canny**. $c^w$'s are the **ground truth** conditions. In the Pose example, the red circle indicates an artifact in the winning example of DPO. In the Canny example, the difference in Canny edge is too hard to discern in raw pixels in DPO, but our method directly compares conditions.

Fig. 2. The cycle consistency is enforced via a loss term between $\mathbf{c}$ and $\hat{\mathbf{c}}$, where $\hat{\mathbf{c}}$ is extracted from the generated image $\hat{\mathbf{x}}$ using a condition detector. For example, in depth-to-image generation, this corresponds to a mean-squared error loss. However, sampling from arbitrary timesteps $t$ with back-propagation is computationally infeasible due to memory constraints. To mitigate this, ControlNet++ employs a *single-step reparameterization* of DDPM [1] to approximate sampling. A key limitation of this approach is the distribution mismatch between the reparameterized diffusion process and sampling processes [1], leading to noisy approximations. As a result, ControlNet++ is only applicable for low-noise steps ($t < 200$), beyond which the sampled image $\hat{\mathbf{x}}$ becomes non-recognizable [11]. As a result, this paradigm fails to optimize high-noise timesteps that are important for image structure generation [12–19].

To avoid noisy approximation and optimize all diffusion timesteps, one approach is to apply *Direct Preference Optimization* for diffusion models (*Diffusion DPO*) [20], to fine-tune a pretrained model. Diffusion DPO computes a *pairwise contrastive loss* at each timestep without requiring an approximation. Let $I$ be an image and $\mathbf{x}_0$ be its corresponding latent representation. Given preference pairs $(I^w, I^l)$ with

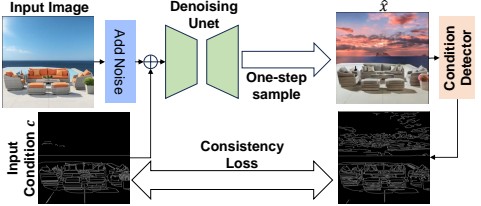

Figure 2: Illustration of ControlNet++.

the same condition $\mathbf{c}$, where $I^w$ is the *winning* image that better aligns with $\mathbf{c}$ (i.e., has a higher controllability score), the model learns to prefer $\mathbf{x}_t^w$ over $\mathbf{x}_t^l$ for any timestep $t$. However, while Diffusion DPO offers general preference optimization capabilities, it faces fundamental limitations for controllable generation tasks. The inherent uncertainty in generative models makes it challenging to create reliable win-lose image pairs that isolate controllability from other quality factors, thus introducing potential noise. To highlight the limitations, we generate a dataset using the pipeline shown in Fig. 1(a), focusing on pose control due to computational constraints[1]. To construct high-quality preference pairs, we generate 20 images per prompt using ControlNet [7] and select the sample with the *highest controllability score* as $I^w$ and the one with the *lowest* as $I^l$. To ensure that $I^w$ is meaningfully better, we use *ImageReward* [21] as a quality filter, requiring $I^w$ to score at least $\Delta = 0.2$ higher than $I^l$. This threshold is sufficiently large that around half of the generated pairs are filtered out. Increasing the threshold further results in too few usable training examples. Despite this setup, several challenges remain for adapting DPO to controllable generation.

**First**, factors unrelated to controllability introduce noise, increasing loss variance and potentially confusing the model. In the pose example shown in Fig. 1(c), although the winning image aligns

---

[1]We generate DPO training data only for Pose, as creating a Canny dataset requires over 5K GPU-days, which is computationally prohibitive. However, we follow the same procedure to generate Canny examples.

better with the pose, it contains a round artifact on the rear of the motorcycle, highlighted in the red circle. Optimizing on such pairs may degrade image quality. Experimental results in Fig. 6 further support this claim.

**Second**, some control types are hard to recognize in raw images. In the Canny edge example shown in Fig. 1(c), differences in Canny edge are hard to discern, while color variations dominate visual perception.

**Third**, the selection process is computationally expensive. Since images are generated using the same model, constructing a preference pair requires multiple inference calls (we use 20) to ensure a diverse range of controllability scores.

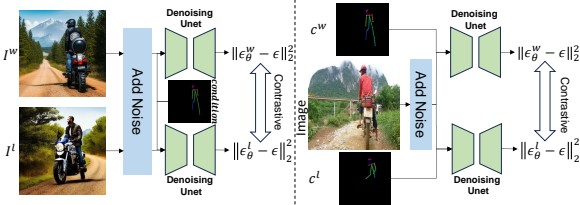

Figure 3: **(a) Training of DPO**. DPO is trained to prefer $I^w$ over $I^l$. **(b) Training of CPO.** CPO is trained to prefer $\mathbf{c}^w$ over $\mathbf{c}^l$.

The first limitation arises because images encode much more information than just alignment with a control signal $\mathbf{c}$. Varying the generated image affects not only its alignment with $\mathbf{c}$, but also unrelated attributes such as background, color, and style. If we could vary only the controllability score while keeping all other aspects fixed, this issue would be mitigated. Importantly, the alignment between an image $I$ and a condition $\mathbf{c}$ is bi-directional. That is, one can compare two images $I^w$ and $I^l$, where $I^w$ aligns better with $\mathbf{c}$, or equivalently, compare two control signals $\mathbf{c}^w$ and $\mathbf{c}^l$, where $\mathbf{c}^w$ aligns better with a fixed image $I$. We adopt the latter formulation by setting $\mathbf{c}^w$ as the **ground-truth** condition and constructing $\mathbf{c}^l$ via perturbation. Specifically, we estimate a control signal from an image generated using $\mathbf{c}^w$ as the condition. In the $(I^w, I^l)$ formulation, the model must infer controllability differences indirectly from raw image content, which is particularly challenging when conditions involve fine-grained details such as Canny edges. In contrast, the $(\mathbf{c}^w, \mathbf{c}^l)$ formulation allows the model to observe controllability differences directly at the condition level, thereby addressing the second limitation. The third limitation of DPO lies in its sampling cost: multiple images must be generated to obtain a meaningful preference pair $(I^w, I^l)$ with a significant difference in controllability. In our approach, this burden is reduced. Since generative models are far from perfectly controllable, the difference between $\mathbf{c}^w$ and its perturbed version $\mathbf{c}^l$ is naturally large enough to serve as a training signal, requiring only a single image sample for perturbation. Based on this insight, we propose **Condition Preference Optimization (CPO)** for controllable image generation by constructing preference pairs $(\mathbf{c}^w, \mathbf{c}^l)$. This method eliminates noise introduced by irrelevant visual factors and provides a cleaner training objective. Our data generation process is illustrated in Fig. 1(b), and the distinction from DPO training is shown in Fig. 3. Our **contributions** are summarized as follows:

- *Low variance training objective than DPO*. Our method provides a low-variance (less noisy) training objective than Diffusion DPO in the control-to-image generation task, and we provide theoretical analysis in Appendix C and experimental support in Sec. 4.2.

- *Generalization beyond ControlNet++*. Our method generalizes beyond ControlNet++ in the sense that we can optimize arbitrary timesteps for improved controllability and support human pose.

- *New Dataset*. We curate and will open-source our Condition Preference (CPO) Dataset. There are 60K examples for human pose, 20K and 118K examples for ADE-20K [22] and COCO-Stuff [23] segmentation, and 2.8M examples each for Canny edges, HED, Lineart, and Depth maps. The CPO dataset is more efficient in curation and storage than the DPO dataset, with details in Appendix D.

- *State-of-the-art Results*. Our method achieves state-of-the-art performance in various tasks. Specifically, we achieve over $10\%$ error rate reduction in segmentation, $70$–$80\%$ error rate reduction in human pose, and around $2$–$5\%$ consistent error rate reduction on edges and depths after fine-tuning. The error rate is defined as the difference between the evaluated controllability and the oracle result.

## 2 Related Works

**Diffusion Models.** Denoising Diffusion Probabilistic Models (DDPM) [1] introduced diffusion models for image generation by employing a forward diffusion process (a Markov chain) that progressively adds noise to an image until it becomes standard Gaussian noise, and a reverse sampling

process that progressively denoises the Gaussian noise to an image using a UNet [24]. However, DDPM discretizes the sampling process into 1000 steps, leading to inefficiencies. To mitigate this, DDIM [25] proposed a non-Markovian formulation that reduces the number of sampling steps without significantly degrading generation quality. Subsequent works [26–28] have developed more advanced ODE solvers to further improve efficiency and sample quality. Beyond sampling steps reduction, other studies have explored leveraging intermediate UNet features to reduce computational cost, such as by reusing feature maps [29] or attention maps [30]. Additionally, latent diffusion models (LDMs) incorporate VQGAN [31] or VAE [32] to project images into a compact latent space for better efficiency. With advancements in model architecture and training scale, LDM is delivered as Stable Diffusion. Building on this foundation, recent works have proposed specialized architectures [7, 10, 9, 8, 33] and diffusion processes [34, 35] tailored to downstream tasks.

**Controllable Image Generation.** Controlling image generation at a fine-grained level, such as aligning with Canny edges or depth maps, is difficult to achieve using text prompts alone, as they offer sparse and coarse control. To address this limitation, ControlNet [7] introduces efficient fine-tuning modules on top of pretrained T2I models (e.g., Stable Diffusion) to incorporate dense visual conditions such as depth and edge maps. UniControl and UniControlNet [10, 9] extend this framework to support multiple control types within a unified architecture. ControlNet++ [11] further improves controllability by enforcing cycle consistency between generated images and input control signals. However, due to the iterative nature of diffusion sampling, it remains infeasible to apply cycle consistency at arbitrary timesteps, as back-propagating through intermediate states requires prohibitive GPU memory [11].

Beyond diffusion-based approaches, ControlAR [36] introduces controllable autoregressive models with multi-resolution training and inference. It employs DINO-v2 [37] as the control signal extractor and achieves competitive performance. However, due to the sequential nature of autoregressive generation, it is significantly less efficient than diffusion models equipped with ODE solvers.

**Direct Preference Optimization.** Direct Preference Optimization (DPO) [38] is a reinforcement learning from human feedback (RLHF) method [39] originally developed for natural language processing. It aims to maximize the log-probability difference between human-preferred and non-preferred responses, encouraging the model to generate preferred outputs. Diffusion-DPO [20] adapts DPO to diffusion models by leveraging a high-quality, human-annotated dataset [40] to enhance image generation quality. Subsequent works introduce alternative optimization techniques, such as KTO [41] and preference score matching [42], to further improve performance [43, 44]. However, in the context of controllable image generation, comparable human-annotated datasets are unavailable. Moreover, applying DPO in this setting presents several limitations, including noisy training objectives, imperceptible control differences (e.g., in edge maps), and high computational overhead, as discussed in Sec. 1.

## 3 Methodology

We first present the preliminaries in Sec. 3.1 and then describe our proposed method in Sec. 3.2.

### 3.1 Preliminaries

**Diffusion Models.** Diffusion models [1] define a forward diffusion process $q(\mathbf{x}_t|\mathbf{x}_0)$ that adds Gaussian noise to an image $\mathbf{x}_0$, and a reverse sampling process $p_\theta(\mathbf{x}_{t-1}|\mathbf{x}_t)$ that reconstructs the image from noisy latent variables. These processes are defined as:

$$q(\mathbf{x}_t|\mathbf{x}_0) = \mathcal{N}(\mathbf{x}_t; \sqrt{\alpha_t}\mathbf{x}_0, (1-\alpha_t)\mathbf{I}), \quad \text{where } \alpha_t = \prod_{s=1}^{t}(1-\beta_s). \tag{1}$$

$$p_\theta(\mathbf{x}_{t-1}|\mathbf{x}_t) = \mathcal{N}(\mathbf{x}_{t-1}; \tilde{\boldsymbol{\mu}}_t, \tilde{\beta}_t), \tag{2}$$

$$\text{where } \tilde{\boldsymbol{\mu}}_t = \frac{1}{1-\beta_t}\left(\mathbf{x}_t - \frac{\beta_t}{\sqrt{1-\alpha_t}}\boldsymbol{\epsilon}\right), \quad \tilde{\beta}_t = \frac{1-\alpha_{t-1}}{1-\alpha_t}\beta_t, \quad \boldsymbol{\epsilon} \sim \mathcal{N}(0, \mathbf{I}). \tag{3}$$

Here, $\beta_t$ is a predefined small constant. In Eq. (3), the noise term $\epsilon$ is unknown and is predicted by a neural network $\epsilon_\theta(\mathbf{x}_t, t)$. Conditioning terms are omitted here for clarity. In practice, they serve as additional inputs to the model.

**Diffusion-DPO.** In the setup of Diffusion-DPO [20], preference pairs $\{\mathbf{x}_0^w, \mathbf{x}_0^l\}$ are provided, and the objective is to train the pretrained model $\epsilon_\theta$ to prefer the winner $\mathbf{x}_0^w$ over the loser $\mathbf{x}_0^l$. Using the Bradley–Terry model [45], the training objective is defined as [20]:

$$\mathcal{L}_{\text{DPO}} = -\mathbb{E}_{\mathbf{c}, \mathbf{x}_0^w, \mathbf{x}_0^l} \log \sigma \left[ \mathbb{E}_{\substack{\mathbf{x}_{1:T}^w \sim p_\theta(\mathbf{x}_{1:T}^w | \mathbf{x}_0^w) \\ \mathbf{x}_{1:T}^l \sim p_\theta(\mathbf{x}_{1:T}^l | \mathbf{x}_0^l)}} \left( \log \frac{p_\theta(\mathbf{x}_0^w | \mathbf{c})}{p_{\text{ref}}(\mathbf{x}_0^w | \mathbf{c})} - \log \frac{p_\theta(\mathbf{x}_0^l | \mathbf{c})}{p_{\text{ref}}(\mathbf{x}_0^l | \mathbf{c})} \right) \right]. \quad (4)$$

With appropriate algebraic and statistical simplifications (see [20] for derivation), this reduces to:

$$\mathcal{L}_{\text{DPO}} = -\mathbb{E}_{(\mathbf{x}_0^w, \mathbf{x}_0^l), t, \epsilon} \Big[ \log \sigma \Big( -\beta T \omega(\lambda_t) \big( \|\epsilon - \epsilon_\theta(\mathbf{x}_t^w, t)\|_2^2 - \|\epsilon - \epsilon_{\text{ref}}(\mathbf{x}_t^w, t)\|_2^2$$
$$- (\|\epsilon - \epsilon_\theta(\mathbf{x}_t^l, t)\|_2^2 - \|\epsilon - \epsilon_{\text{ref}}(\mathbf{x}_t^l, t)\|_2^2) \big) \Big) \Big]. \quad (5)$$

Here, $\epsilon$ is sampled from $\mathcal{N}(0, \mathbf{I})$, and $\mathbf{x}_t^w$ and $\mathbf{x}_t^l$ are obtained via Eq. (1). The networks $\epsilon_\theta$ and $\epsilon_{\text{ref}}$ are initialized with the same weights, but only $\epsilon_\theta$ is updated during training. The scalar $\lambda_t$ is defined as $\frac{\alpha_t}{1 - \alpha_t}$, and $\omega(\lambda_t)$ is practically set to 0.5 [20].

## 3.2 Proposed Condition Preference Optimization

**Formulation.** For notational simplicity, we omit the text prompt. Given a triplet $(\mathbf{x}_0, \mathbf{c}^w, \mathbf{c}^l)$, where $\mathbf{x}_0$ is the latent representation of an image $I$, and $\mathbf{c}^w$, $\mathbf{c}^l$ are control signals (e.g., depth maps), we have $\mathbf{c}^w$ aligning better with $I$ than $\mathbf{c}^l$. Our Condition Preference Optimization (CPO) loss is:

$$\mathcal{L}_{\text{CPO}} = -\mathbb{E}_{\mathbf{c}^w, \mathbf{c}^l, \mathbf{x}_0} \log \sigma \left[ \mathbb{E}_{\mathbf{x}_{1:T} \sim p_\theta(\mathbf{x}_{1:T} | \mathbf{x}_0)} \left( \log \frac{p_\theta(\mathbf{x}_0 | \mathbf{c}^w)}{p_{\text{ref}}(\mathbf{x}_0 | \mathbf{c}^w)} - \log \frac{p_\theta(\mathbf{x}_0 | \mathbf{c}^l)}{p_{\text{ref}}(\mathbf{x}_0 | \mathbf{c}^l)} \right) \right]. \quad (6)$$

Similar to Eq. (5), the expression can be simplified (full derivation is provided in Appendix B) as:

$$\mathcal{L}_{\text{CPO}} = -\mathbb{E}_{(\mathbf{c}^w, \mathbf{c}^l, \mathbf{x}_0), t, \epsilon} \Big[ \log \sigma \Big( -\beta T \omega(\lambda_t) \big( \|\epsilon - \epsilon_\theta(\mathbf{x}_t, \mathbf{c}^w, t)\|_2^2 - \|\epsilon - \epsilon_{\text{ref}}(\mathbf{x}_t, \mathbf{c}^w, t)\|_2^2$$
$$- (\|\epsilon - \epsilon_\theta(\mathbf{x}_t, \mathbf{c}^l, t)\|_2^2 - \|\epsilon - \epsilon_{\text{ref}}(\mathbf{x}_t, \mathbf{c}^l, t)\|_2^2) \big) \Big) \Big]. \quad (7)$$

**Gradient Analysis for the Improved Formulation.** We begin by defining the following terms:

$$\alpha = \beta T \omega, \quad d_\theta = \|\epsilon - \epsilon_\theta(\mathbf{x}_t, \mathbf{c}^w, t)\|_2^2 - \|\epsilon - \epsilon_\theta(\mathbf{x}_t, \mathbf{c}^l, t)\|_2^2, \quad d_{\text{ref}} = \|\epsilon - \epsilon_{\text{ref}}(\mathbf{x}_t, \mathbf{c}^w, t)\|_2^2 - \|\epsilon - \epsilon_{\text{ref}}(\mathbf{x}_t, \mathbf{c}^l, t)\|_2^2. \quad (8)$$

The gradient of $\mathcal{L}_{\text{CPO}}$ (ignoring the expectation for clarity) is:

$$\nabla_\theta \mathcal{L}_{\text{CPO}} = -\sigma \left( \alpha \cdot (d_\theta - d_{\text{ref}}) \right) \cdot \nabla_\theta \left( -\alpha \cdot (d_\theta - d_{\text{ref}}) \right)$$
$$= \alpha \cdot \sigma \left( \alpha \cdot (d_\theta - d_{\text{ref}}) \right) \cdot \nabla_\theta d_\theta. \quad (9)$$

This is equivalent to scaling the gradient of a loss defined by $d_\theta$. The term $d_\theta$ resembles a triplet loss, where $\epsilon$ serves as the *anchor*, $\epsilon_\theta(\mathbf{x}_t, \mathbf{c}^w, t)$ as the *positive sample*, and $\epsilon_\theta(\mathbf{x}_t, \mathbf{c}^l, t)$ as the *negative sample*, but without a margin term and zero-clipping. The margin term and zero-clipping are introduced to prevent excessive contrast between positive and negative pairs.

Importantly, in CPO, the negative sample does not imply "incorrectness", but rather a weak alignment between image and condition. Therefore, we include a triplet margin to avoid overly contrasting the two samples. In addition, since $\alpha$ is typically set to a large value (e.g., 2500) [20], it can amplify the gradient and destabilize training. To mitigate this, we remove the scaling factor $\alpha$ from the outside.

Our final CPO loss is formulated as:

$$\mathcal{L}_{\text{CPO}} = \mathbb{E}_{(\mathbf{c}^w, \mathbf{c}^l, \mathbf{x}_0), t, \epsilon} \left[ \lambda_{\text{CPO}} \cdot \max(d_\theta + m, 0) \right],$$
$$\text{where } \lambda_{\text{CPO}} = \text{sg} \left( \alpha \cdot (d_\theta - d_{\text{ref}}) \right), \text{ and } d_* = \|\epsilon - \epsilon_*(\mathbf{x}_t, \mathbf{c}^w, t)\|_2^2 - \|\epsilon - \epsilon_*(\mathbf{x}_t, \mathbf{c}^l, t)\|_2^2. \quad (10)$$

Table 1: Quantitative comparison in controllability. ↑ means higher is better, ↓ means lower is better. The best results are **boldfaced**. **Oracle** indicates the performance of the condition detection model on ground truth images, which is the **upper-bound** of controllability. We only compare models with the same CFG Sales for fairness. - No publicly available checkpoints or evaluation results are available. †Models are not open-sourced, but results are copied from their papers. * Evaluated with their provided checkpoints.

| Method | T2I Model | Segmentation | | Pose | | Canny | HED | Lineart | Depth |
|---|---|---|---|---|---|---|---|---|---|
| | | ADE20K | COCO-Stuff | COCO-Pose | HumanArt | | MultiGen-20M | | |
| | | mIoU ↑ | | mAP ↑ | | F1 ↑ | SSIM ↑ | SSIM ↑ | RMSE ↓ |
| ControlAR (CFG = 4.0) | LlamaGen-XL | 39.95† | **37.49** | - | - | 36.78* | 0.8184* | 0.7922† | 29.33* |
| Ours (CFG = 4.0) | SD1.5 | **46.38** | 36.10 | - | - | **39.68** | **0.8299** | **0.8559** | **25.98** |
| T2I-Adapter | SD1.5 | 12.61 | - | - | - | 23.65 | - | - | 48.40 |
| Gligen | SD1.4 | 23.78 | - | 87.84 | 39.88 | 26.94 | 0.5634 | - | 38.83 |
| Uni-ControlNet | SD1.5 | 19.39 | - | 12.50 | 4.23 | 27.32 | 0.6910 | - | 40.65 |
| UniControl | SD1.5 | 25.44 | - | 75.29 | 27.90 | 30.82 | 0.7969 | - | 39.18 |
| ControlNet | SD1.5 | 32.55 | 27.46 | 72.53 | 34.52 | 34.65 | 0.7621 | 0.7054 | 35.90 |
| ControlNet++ | SD1.5 | 43.64 | 34.56 | - | - | 38.03* | 0.8097 | 0.8399 | 28.32 |
| Ours | SD1.5 | **44.81** | **35.49** | **87.98** | **45.71** | **39.28** | **0.8201** | **0.8447** | **27.49** |
| Oracle | - | 55.07 | 41.32 | 92.54 | 50.03 | 100.0 | 1.0 | 1.0 | 0.0 |

Here, sg$(\cdot)$ denotes the stop-gradient operation. Since $d_\theta$ diverges from the original diffusion model objective, we add the pretrained diffusion loss as a regularization term. The final loss becomes:

$$\mathcal{L}_{\text{total}} = \mathcal{L}_{\text{CPO}} + \lambda\mathcal{L}_{\text{pretrain}}, \quad \text{where} \quad \mathcal{L}_{\text{pretrain}} = \mathbb{E}_{(\mathbf{x}_0, \mathbf{c}^w), \, t', \, \epsilon'} \left\| \epsilon' - \epsilon_\theta(\mathbf{x}_{t'}, \mathbf{c}^w, t') \right\|_2^2. \quad (11)$$

The variables $\epsilon'$ and $t'$ denote independently resampled noise and timesteps. A proof that the variance of the CPO loss is lower than that of DPO (i.e., it is less noisy) is provided in Appendix C.

**CPO Dataset Curation.** Suppose we have a detection model $R$ that extracts control signals from images, a generative model $G$ that generates images conditioned on control signals, and a dataset consisting of images $I$ and control signals $\mathbf{c}$ (text prompts are omitted for simplicity). Our data curation process proceeds as follows: **(1)** Generate an image $\hat{I}$ using $G$ conditioned on $\mathbf{c}$; **(2)** Set the winning condition $\mathbf{c}^w = \mathbf{c}$ and obtain the losing condition $\mathbf{c}^l = R(\hat{I})$; **(3)** If the original control $\mathbf{c}$ is unavailable, fallback to $\mathbf{c}^w = R(I)$; **(4)** Store the triplet $(I, \mathbf{c}^w, \mathbf{c}^l)$.

We select $G$ to be ControlNet++ when available; otherwise, we use ControlNet. A detailed comparison with DPO dataset curation is provided in Appendix D.

## 4   Experiments

**Datasets and Curation.** Following ControlNet++ [11], we use ADE20K [22] and COCO-Stuff [23] for segmentation-to-image generation, and MultiGen-20M [10] for edge and structure-based conditions, including Canny edges, HED, Lineart, and depth maps. In addition, we include COCO-Pose [46] and HumanArt [47] for pose-to-image generation. Note that we only train the pose-to-image generation model on COCO-Pose and test it on both COCO-Pose and HumanArt. We curate CPO datasets using COCO-Pose, ADE20K, COCO-Stuff, and MultiGen-20M. For COCO-Pose, we generate images using ControlNet [7] and apply YOLO-11x-Pose [48] to extract keypoints from the generated images, which serve as the perturbed condition $\mathbf{c}^l$. For the remaining datasets, we use ControlNet++ to generate images and apply the same condition detectors as in ControlNet++.

**Evaluation and Metrics.** Following ControlNet++ [11], we resize all images to $512\times512$ for evaluation. We use mean Intersection over Union (mIoU) for segmentation, root mean squared error (RMSE) for depth, and structural similarity index (SSIM) for Lineart and HED. For Canny edges, we report the F1 score, and for human pose, we follow conventions in pose estimation [49] to report mean Average Precision (mAP). To ensure fair comparison, we use the same condition detectors as ControlNet++ to extract conditions from the generated images. For pose estimation, we use YOLO-11x-Pose [48] to detect human keypoints. In addition to controllability metrics, we report Fréchet Inception Distance (FID) using the CleanFID implementation [50] and CLIP similarity scores [51]. For consistency across all experiments, we fix the classifier-free guidance scale to 7.5 (the standard for SD1.5-based models) and use the UniPC sampler [28] with 20 inference steps, matching the ControlNet++ setup. Training details are provided in Appendix E.

**Baselines and Base Models.** We compare against several strong controllable generation baselines, including T2I-Adapter [8], GLIGEN [33], Uni-ControlNet [9], UniControl [10], ControlNet [7],

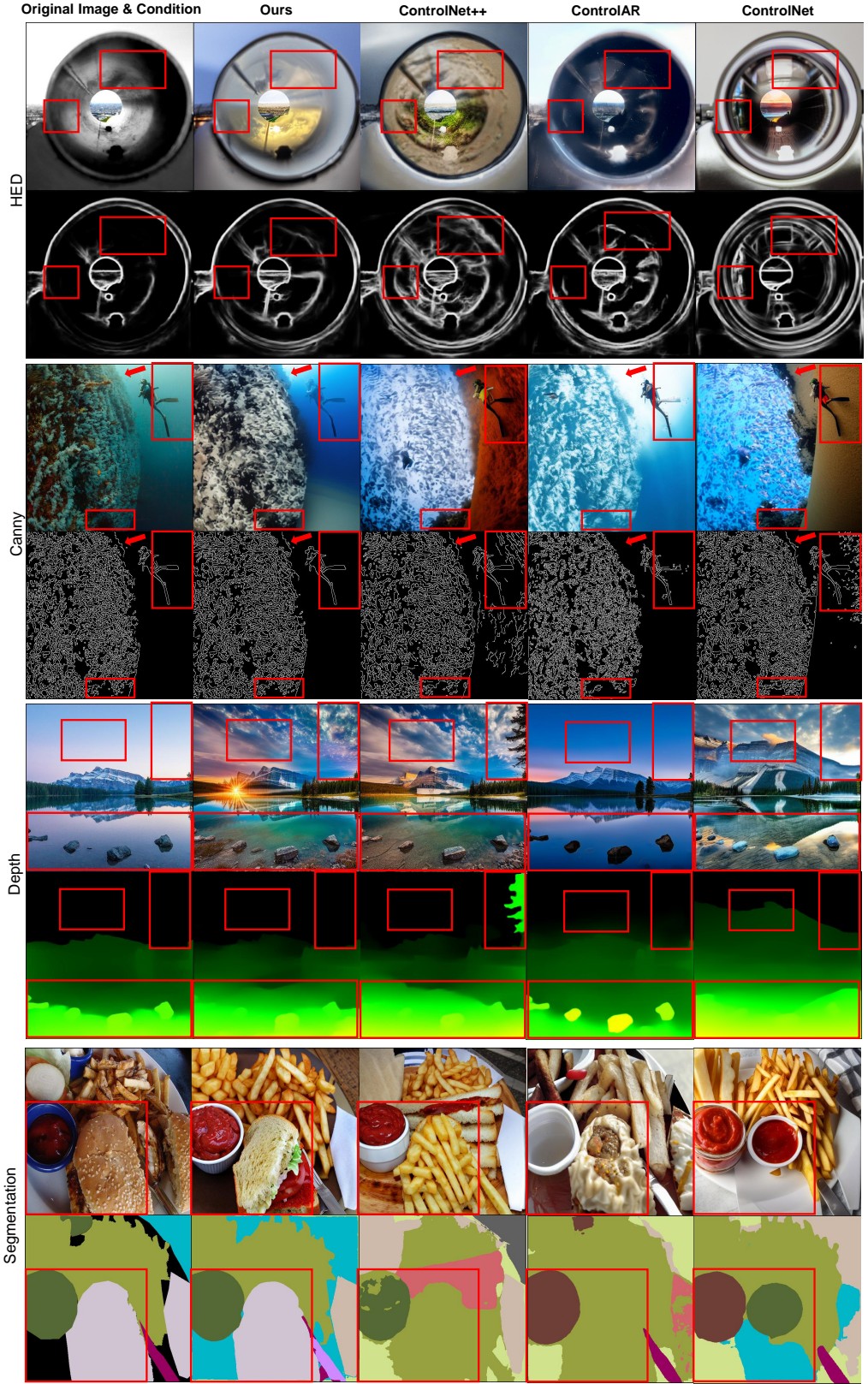

Figure 4: Qualitative Comparison in Controllability. Red boxes indicate the area where our method achieves better controllability.

Table 2: Results in FID↓/CLIP↑. Best results are **boldfaced**. We only compare models with the same CFG scales.

| Method | T2I Model | Segmentation | | Pose | | Canny | HED | Lineart | Depth |
|---|---|---|---|---|---|---|---|---|---|
| | | ADE20K | COCO-Stuff | COCO-Pose | HumanArt | | MultiGen-20M | | |
| ControlAR (CFG = 4.0) | LlamaGen-XL | **27.15**†/ - | **14.51**/31.09* | - | - | 19.00*/29.12* | 14.03*/30.82* | **12.41**†/ - | 17.70*/29.19* |
| Ours (CFG = 4.0) | SD1.5 | 27.59/**31.80** | 17.01/**32.20** | - | - | **18.49/31.15** | **12.24/31.55** | 12.98/**31.44** | **14.62/31.72** |
| T2I-Adapter | SD1.5 | 39.15/30.65 | - | - | - | 15.96/31.71 | - | - | 22.52/31.46 |
| Gligen | SD1.4 | 33.02/31.12 | - | 41.36/32.13 | 48.92/31.20 | 18.89*/31.77 | - | - | 18.36/31.75 |
| Uni-ControlNet | SD1.5 | 30.97/30.59 | - | 46.19/31.78 | 37.26/33.50 | 17.14/31.84 | 17.08/31.94 | - | 20.27/31.66 |
| UniControl | SD1.5 | 46.34/30.92 | - | 44.74/31.92 | 46.39/33.28 | 19.94/31.97 | 15.99/32.02 | - | 18.66/**32.45** |
| ControlNet | SD1.5 | 33.28/31.53 | 21.33/32.21* | 42.02/32.05 | **32.22/33.83** | **14.73/32.15** | 15.47/**32.33** | 17.44/**32.46** | 17.76/**32.45** |
| ControlNet++ | SD1.5 | **30.24***/31.96 | 19.79*/32.25* | - | - | 20.16*/31.87 | 15.01/32.05 | 13.88/31.95 | **16.66**/32.17 |
| Ours | SD1.5 | 30.30/**31.97** | **19.30/32.36** | **39.21/32.90** | 39.94/32.98 | 19.69/31.83 | **13.35**/32.07 | **13.35**/31.98 | **15.88**/32.31 |

ControlAR [36], and ControlNet++ [11]. As our method is designed as a fine-tuning approach, it requires initialization from a pretrained model. We use ControlNet as the base model for human pose generation and ControlNet++ for all other tasks.

## 4.1 Experimental Results

**Fair Evaluation.** To ensure fair comparison, we re-evaluate the results of ControlNet++ and ControlAR using their official checkpoints. If substantial differences are observed due to variations in the machine environment or implementation details, we report our re-evaluated results instead. Specifically, ControlNet-based methods use the Kornia implementation for Canny edge detection, whereas ControlAR uses an OpenCV-based implementation with a different threshold, which can lead to inconsistencies. Additionally, there are subtle differences in depth and HED preprocessing: ControlNet-based methods round these maps to integers during image generation and evaluation, while ControlAR retains floating-point values. All re-evaluated results are marked with *. Furthermore, ControlNet-based methods are evaluated with a classifier-free guidance (CFG) scale of 7.5 [52], while ControlAR is evaluated at CFG scale 4.0. To ensure fairness, we also evaluate our method at CFG scale 4.0 and only compare methods under the same CFG scale.

**Controllability Results.** Controllability evaluation results are presented in Tab. 1. In the MultiGen-20M dataset, labels are generated using the same condition detectors employed for evaluation, enabling oracle controllability scores to be perfect when extracting conditions from ground-truth images. In contrast, labels for human pose and segmentation tasks are human-annotated, making evaluation upper-bounded by the accuracy of the condition detector. The error rate is defined as the difference between the evaluated controllability score and the oracle result. Our method consistently yields substantial improvements over the base models it fine-tunes. Specifically, we observe approximately 10% error rate reduction on ADE20K, 14% on COCO-Stuff, and over 70% on both pose datasets. For edge-based control types, we

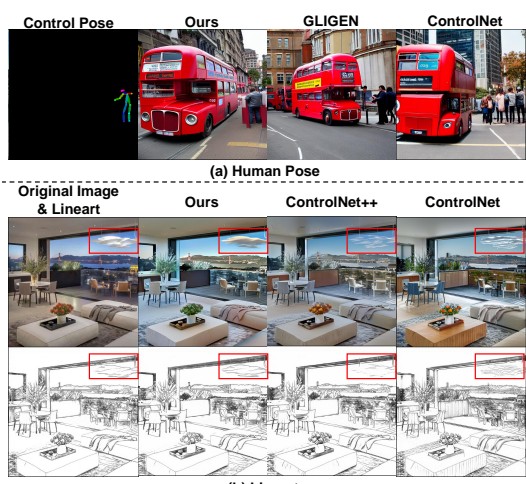

(a) Human Pose

(b) Lineart

Figure 5: Qualitative Comparison in Pose and Lineart.

observe a 2% error rate reduction on Canny and Lineart, 5% on HED, and 3% on depth. The relatively modest improvements for fine-grained signals like Canny and Lineart may be due to the sparsity of supervision in the latent space. In contrast, HED captures coarser structural cues and thus enables more noticeable gains. For object-level control conditions such as human pose and segmentation, our method achieves significantly more pronounced improvements. Compared to ControlAR, our approach demonstrates substantially stronger controllability performance across most settings, except for COCO-Stuff.

**FID and CLIP.** We report FID and CLIP scores in Tab. 2. Compared to ControlAR, our method achieves comparable FID scores while significantly outperforming it in CLIP scores. Relative to ControlNet++, our method has minimal impact on both metrics, suggesting that the controllability improvements from fine-tuning do not compromise overall generation quality or semantic alignment. We note that our FID score on HumanArt is worse than that of the base model, ControlNet. This

Table 3: User study on controllability.

| Task | CPO | ControlNet++ | ControlAR |
|------|-----|--------------|-----------|
| HED | 47.73% | 37.87% | 14.40% |
| Depth | 46.67% | 40.27% | 13.06% |

Table 4: Ablation studies on the effect of DINO-v2 feature extraction on controllability. ↑ means higher is better, ↓ means lower is better. The best results are **boldfaced**. **Oracle** indicates the performance of the condition detection model on ground truth images, which is the **upper-bound** of controllability.

| Method | T2I Model | Segmentation | | Pose | | Canny | HED | Lineart | Depth |
|--------|-----------|--------------|---|------|---|-------|-----|---------|-------|
| | | ADE20K | COCO-Stuff | COCO-Pose | HumanArt | \multicolumn{4}{c}{MultiGen-20M} | | | |
| | | mIoU ↑ | | mAP ↑ | | F1 ↑ | SSIM ↑ | SSIM ↑ | RMSE ↓ |
| Ours | SD1.5 | **44.81** | 35.49 | 87.98 | 45.71 | 39.28 | 0.8201 | **0.8447** | 27.49 |
| Ours-DINOv2 | SD1.5 | 43.72 | **39.25** | **88.45** | **49.98** | **44.81** | **0.8376** | 0.8301 | **26.92** |
| Oracle | - | 55.07 | 41.32 | 92.54 | 50.03 | 100.0 | 1.0 | 1.0 | 0.0 |

is likely due to a domain gap: HumanArt consists predominantly of cartoon-style images, while our training data (COCO-Pose) comprises real-world photographs. A similar pattern is observed with GLIGEN, which is also trained on COCO-Pose. However, our method generalizes better than GLIGEN on this out-of-distribution dataset, indicating stronger robustness despite the domain shift. Finally, we observe that the classifier-free guidance (CFG) scale significantly affects both controllability and FID. We include an ablation study on this in Sec. 4.2.

**Qualitative Comparison.** We present qualitative comparisons with baseline methods in Figs. 4 and 5. For HED and Canny edge conditions, other methods produce undesired edge artifacts within the highlighted red boxes, while ControlAR additionally misses key edge segments, indicated by the red arrows. In the depth example, the original image depicts a shallow river near the camera, where the riverbed is visible. ControlAR fails to capture this detail, ControlNet++ introduces an undesired tree, and ControlNet generates an over-height mountain. In the segmentation example, ControlAR fails to generate the ketchup and the sandwich, while both ControlNet++ and ControlNet omit the sandwich. Notably, our method is the only one that correctly generates the table in the upper-right corner. The black region in the input condition represents background and is not evaluated for correctness. In the pose example (Fig. 5), both GLIGEN and ControlNet mistakenly generate multiple humans, whereas our method accurately renders a single person. In the Lineart case, our method produces more coherent results in generating the cloud-shaped object highlighted in the red box.

**User Study.** We conduct a user study on Controllability among our methods, ControlNet++, and ControlAR. We randomly sample 25 examples each from HED and depth and ask 15 annotators to select which one is the most consistent with the input condition. The results in win rate (%) are shown in Tab. 3.

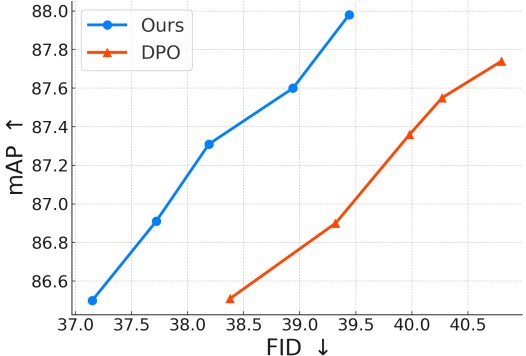

Figure 6: Comparison with DPO. We evaluated 5 checkpoints equally throughout training and observed that under the same mAP level, our method gets better FID.

## 4.2 Ablation Study

**CPO vs. DPO.** Since DPO requires additional dataset curation, we provide a quantitative comparison on COCO-Pose [46] only. We evaluate five checkpoints evenly spaced throughout training and report results in Fig. 6. Our method consistently achieves higher mAP at matched FID levels, or lower FID at matched mAP levels, demonstrating improved controllability–quality trade-offs over DPO.

**DINO-v2 adaptation.** ControlAR adopts pretrained DINO-v2 [37] to extract features of conditions, while ControlNet-based methods use simple CNNs. The DINO-v2 is also adaptable to ControlNet-based methods and generally improves image quality and controllability, shown in Tab. 4 and 5. *Ours-DINOv2* is obtained by pretraining a ControlNet with the DINO-v2 adapter until convergence, followed by our CPO finetuning for at most 4000 steps.[2]

---

[2]We train ControlNet++ for COCOStuff and Depth for 500 steps and 2000 steps, respectively. For other tasks, we do not train ControlNet++.

Table 5: Ablation studies on the effect of DINO-v2 feature extraction on FID↓/CLIP↑. Best results are **boldfaced**.

| Method | T2I Model | Segmentation | | Pose | | Canny | HED | Lineart | Depth |
|---|---|---|---|---|---|---|---|---|---|
| | | ADE20K | COCO-Stuff | COCO-Pose | HumanArt | | MultiGen-20M | | |
| Ours | SD1.5 | 30.30/31.97 | 19.30/32.36 | 39.21/**32.90** | **39.94/32.98** | 19.69/31.83 | 13.35/32.07 | 13.35/31.98 | 15.88/32.31 |
| Ours-DINOv2 | SD1.5 | **30.02/32.02** | **17.57/32.53** | **37.58**/32.73 | 43.78/32.45 | **14.62/32.07** | **12.06/32.07** | **13.12/32.03** | **15.50/32.45** |

Table 6: Ablation Studies on CFG Scales.

| CFG scale | Seg. (ADE-20K) | Canny |
|---|---|---|
| | mIoU↑/FID↓/CLIP↑ | F1↑/FID↓/CLIP↑ |
| 1.5 | 45.18/28.28/30.95 | 39.49/20.45/29.54 |
| 3.0 | **46.40**/27.21/31.68 | **39.69**/18.95/30.75 |
| 4.0 | 46.38/27.59/31.80 | 39.68/**18.49**/31.15 |
| 7.5 | 44.81/30.30/31.97 | 39.28/19.69/**31.83** |
| ControlAR (4.0) | 39.95/**27.15**/- | 36.78/19.00/29.12 |

Table 7: Ablation Studies of margin and regularization strength on ADE-20K dataset.

| Margin | Reg. Strength | mIoU↑/FID↓/CLIP↑ |
|---|---|---|
| $m = 0.01$ | $\lambda = 0.05$ | 44.81/30.30/31.97 |
| $m = 0.1$ | $\lambda = 0.05$ | 44.81/30.47/31.96 |
| no margin | $\lambda = 0.05$ | 44.83/30.79/31.93 |
| $m = 0.01$ | $\lambda = 0.15$ | 44.09/29.98/32.00 |
| $m = 0.01$ | $\lambda = 0$ | 46.01/31.37/31.92 |
| $m = 0.01$ | $\lambda = 1$ | 39.01/30.80/31.77 |

**Classifier-Free Guidance.** We conduct experiments on varying the classifier-free guidance (CFG) scale for our method in Tab. 6. Additional results are in Appendix F. We observe that increasing the CFG scale beyond a certain point (around 3–4) leads to a degradation in both FID and controllability, while CLIP scores continue to improve. This trade-off between FID and CLIP scores has also been observed in prior work [2], and our results show a similar trend extending to controllability. These findings highlight an open research question: **how can we fairly evaluate controllable image generation**, given the inherent trade-offs among controllability, FID, and CLIP, all of which are influenced by the CFG scale.

**Regularization Strength and Margin.** We conduct an ablation study on the margin $m$ and regularization strength $\lambda$ in Eqs. (9) and (10) to evaluate the effect of each component in our design. Results are reported in Tab. 7. In the first three rows, we fix $\lambda$ and vary the margin $m$. In the last three rows, we fix $m$ and vary $\lambda$. The case of "no margin" corresponds to using an untruncated contrastive loss. To better interpret the results, we evaluate checkpoints that achieve similar mIoU when varying $m$. For the regularization strength $\lambda$, models are trained for 10K steps. Interestingly, when $\lambda$ is set too high, performance deteriorates. This may be due to the pretraining loss dominating the CPO objective, causing the model to revert toward the original ControlNet behavior. When the regularization term is removed (i.e., $\lambda = 0$), controllability improves substantially, reducing the error rate by around $10\%$, but both FID and CLIP scores degrade. At $\lambda = 0.15$, controllability gains are smaller, but FID and CLIP scores improve. Notably, although models trained without regularization ($\lambda = 0$) achieve worse FID, the drop in visual quality is not always perceptible to humans, as shown in our Appendix F.

## 5  Conclusion

In this paper, we propose Condition Preference Optimization (CPO), a low-variance training objective compared to DPO. CPO achieves better FID scores at matched Controllability, benefiting from reduced variance and a disentangled training signal. In addition, CPO supports training at arbitrary timesteps, allowing it to generalize beyond ControlNet++ and achieve state-of-the-art controllability across a variety of tasks. Through evaluation across different classifier-free guidance (CFG) scales, we also uncover a broader challenge in the fair evaluation of controllable image generation: specifically, how to jointly assess controllability, FID, and CLIP scores.

## 6  Acknowledgment

This work was supported by the intramural research program of the U.S. Department of Agriculture, National Institute of Food and Agriculture via grant number 2024-67022-41788. Any opinions, findings, conclusions, or recommendations expressed in this publication are those of the author(s) and should not be construed to represent any official USDA or U.S. Government determination or policy.

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

# Appendix

Our appendix is structured as follows:

- A: Limitations and Broader Impact.
- B: Full derivation of the CPO loss.
- C: Proof of the claim that the CPO loss has lower variance than DPO.
- D: Comparison between the CPO and DPO datasets, including analysis of computation and storage costs.
- E: Training details, including hyperparameters and hardware specifications.
- F: Additional ablation studies.
- G: Additional visual results and selected failure cases.

## A  Limitations and Broader Impact

**Limitation and Future Work.** Recent works such as ControlAR [36] enable multi-resolution training and inference, while ControlNet-based methods (using Stable Diffusion v1.5 as the base T2I model) are typically trained and evaluated at a fixed $512 \times 512$ resolution. Additionally, it may be interesting to apply our method to other RLHF algorithms such as KTO [41].

**Broader Impact.** As a controllable generative model, our method may have a positive societal impact in areas such as privacy preservation—for example, generating images conditioned on Lineart extracted from humans to obscure identity. However, similar to general-purpose generative models, it carries risks of malicious misuse, including the generation of unsafe or inappropriate content (e.g., nudity).

## B  Full Derivation of CPO loss

Our Condition Preference Optimization (CPO) loss is defined as:

$$\mathcal{L}_{CPO} = -\mathbb{E}_{\mathbf{c}^w, \mathbf{c}^l, \mathbf{x}_0} \log \sigma \left[ \mathbb{E}_{\mathbf{x}_{1:T} \sim p_\theta(\mathbf{x}_{1:T}|\mathbf{x}_0)} \left( \log \frac{p_\theta(\mathbf{x}_0|\mathbf{c}^w)}{p_{\text{ref}}(\mathbf{x}_0|\mathbf{c}^w)} - \log \frac{p_\theta(\mathbf{x}_0|\mathbf{c}^l)}{p_{\text{ref}}(\mathbf{x}_0|\mathbf{c}^l)} \right) \right]. \quad (12)$$

$\sigma$ is the sigmoid function. Since sampling from $p_\theta$ is intractable, we approximate it using the diffusion process defined in Eq. (1). Because the *log-sigmoid* function is convex, we can apply Jensen's inequality to move the expectation outside, resulting in the following (condition $\mathbf{c}$ omitted for notational simplicity):

$$\begin{aligned}
\mathcal{L}_{CPO} \leq -\mathbb{E}_{(\mathbf{x}_0, \mathbf{c}^w, \mathbf{c}^l), t} \Big[ \log \sigma \Big( - \beta T \big( &+ \mathbb{D}_{\text{KL}}(q(\mathbf{x}_{t-1} \mid \mathbf{x}_0, t) \, \| \, p_\theta(\mathbf{x}_{t-1} \mid \mathbf{x}_t, \mathbf{c}^w)) \\
&- \mathbb{D}_{\text{KL}}(q(\mathbf{x}_{t-1} \mid \mathbf{x}_0, t) \, \| \, p_{\text{ref}}(\mathbf{x}_{t-1} \mid \mathbf{x}_t, \mathbf{c}^w)) \\
&- \mathbb{D}_{\text{KL}}(q(\mathbf{x}_{t-1} \mid \mathbf{x}_0, t) \, \| \, p_\theta(\mathbf{x}_{t-1} \mid \mathbf{x}_t, \mathbf{c}^l)) \\
&+ \mathbb{D}_{\text{KL}}(q(\mathbf{x}_{t-1} \mid \mathbf{x}_0, t) \, \| \, p_{\text{ref}}(\mathbf{x}_{t-1} \mid \mathbf{x}_t, \mathbf{c}^l))) \big) \Big) \Big]
\end{aligned} \quad (13)$$

By [1], each KL divergence term can be simplified, yielding the following final form:

$$\begin{aligned}
\mathcal{L}_{CPO} = -\mathbb{E}_{(\mathbf{c}^w, \mathbf{c}^l, \mathbf{x}_0), t, \epsilon} \Big[ \log \sigma \Big( - \beta T \omega(\lambda_t) \Big( \| \epsilon - \epsilon_\theta(\mathbf{x}_t, \mathbf{c}^w, t) \|_2^2 - \| \epsilon - \epsilon_{\text{ref}}(\mathbf{x}_t, \mathbf{c}^w, t) \|_2^2 \\
- \big( \| \epsilon - \epsilon_\theta(\mathbf{x}_t, \mathbf{c}^l, t) \|_2^2 - \| \epsilon - \epsilon_{\text{ref}}(\mathbf{x}_t, \mathbf{c}^l, t) \|_2^2 \big) \Big) \Big) \Big].
\end{aligned} \quad (14)$$

## C  Proof of Loss Variance

We include the proof that our CPO loss achieves lower variance here.

**1. Decomposition of inputs.** We write each of the two inputs $u^+$ ("winning") and $u^-$ ("losing") as

$$u^\pm = \bar{u} + \delta_{\text{ctrl}}^\pm + \delta_{\text{nuis}}^\pm. \tag{15}$$

For example, $u^+$ is $\mathbf{x}_t^w$, $\mathbf{c}$ for DPO and $\mathbf{x}_t$, $\mathbf{c}^w$ for CPO. $u^-$ is defined similarly for the losing example. $\bar{u}$ is a shared baseline, specifically original input $\mathbf{x}_t$, $\mathbf{c}$, $\delta_{\text{ctrl}}$ is the deviation due to condition alignment, and $\delta_{\text{nuis}}$ collects all other deviations. Define

$$\Delta_{\text{ctrl}} = \delta_{\text{ctrl}}^+ - \delta_{\text{ctrl}}^-, \quad \Delta_{\text{nuis}} = \delta_{\text{nuis}}^+ - \delta_{\text{nuis}}^-, \quad u^+ - u^- = \Delta_{\text{ctrl}} + \Delta_{\text{nuis}}. \tag{16}$$

**2. Centering nonzero means.** If the deviations have nonzero mean, set

$$\mu_{\text{ctrl}} = \mathbb{E}[\Delta_{\text{ctrl}}], \quad \mu_{\text{nuis}} = \mathbb{E}[\Delta_{\text{nuis}}], \tag{17}$$

and define residuals

$$\widetilde{\Delta}_{\text{ctrl}} = \Delta_{\text{ctrl}} - \mu_{\text{ctrl}}, \quad \widetilde{\Delta}_{\text{nuis}} = \Delta_{\text{nuis}} - \mu_{\text{nuis}}. \tag{18}$$

**3. Linear approximation of the score difference.** We first define:

$$s_\theta(u) = -||\epsilon_\theta(u) - \epsilon||_2^2 = -||\epsilon_\theta(\mathbf{x}_t, \mathbf{c}) - \epsilon||_2^2. \tag{19}$$

We ignore input $t$ as it is not relevant. Let $g = \nabla_u s_\theta(\bar{u})$. By a first-order Taylor expansion,

$$\Delta s = s_\theta(u^+) - s_\theta(u^-) \approx \langle g, \ u^+ - u^- \rangle = \langle g, \ \mu_{\text{ctrl}} + \mu_{\text{nuis}} \rangle + \langle g, \ \widetilde{\Delta}_{\text{ctrl}} + \widetilde{\Delta}_{\text{nuis}} \rangle. \tag{20}$$

**4. Variance decomposition.** The variance of $\Delta s$ is

$$\text{Var}[\Delta s] = \mathbb{E}\Big[\big(\langle g, Z \rangle - \mathbb{E}[\langle g, Z \rangle]\big)^2\Big] \tag{21}$$

$$= \mathbb{E}\big[\langle g, Z \rangle^2\big] - \big(\mathbb{E}[\langle g, Z \rangle]\big)^2 \tag{22}$$

$$= \mathbb{E}\Big[\big(g^\top(\widetilde{\Delta}_{\text{ctrl}} + \widetilde{\Delta}_{\text{nuis}})\big)^2\Big] - \big(g^\top \mathbb{E}[\widetilde{\Delta}_{\text{ctrl}} + \widetilde{\Delta}_{\text{nuis}}]\big)^2 \tag{23}$$

$$= g^\top \mathbb{E}\big[(\widetilde{\Delta}_{\text{ctrl}} + \widetilde{\Delta}_{\text{nuis}})(\widetilde{\Delta}_{\text{ctrl}} + \widetilde{\Delta}_{\text{nuis}})^\top\big] g - g^\top \underbrace{\big(\mathbb{E}[\widetilde{\Delta}_{\text{ctrl}} + \widetilde{\Delta}_{\text{nuis}}]\big)\big(\mathbb{E}[\widetilde{\Delta}_{\text{ctrl}} + \widetilde{\Delta}_{\text{nuis}}]\big)^\top}_{=0} g \tag{24}$$

$$\text{Var}[\Delta s] = g^\top \Big[\underbrace{\text{Cov}(\widetilde{\Delta}_{\text{ctrl}})}_{V_{\text{ctrl}}} + \underbrace{\text{Cov}(\widetilde{\Delta}_{\text{nuis}})}_{V_{\text{nuis}}} + 2\underbrace{\text{Cov}(\widetilde{\Delta}_{\text{ctrl}}, \widetilde{\Delta}_{\text{nuis}})}_{V_{\text{cross}}}\Big] g. \tag{25}$$

**5. DPO case.** Due to the inherent randomness of sampling process, both $\Delta_{\text{ctrl}}$ and $\Delta_{\text{nuis}}$ vary, giving

$$V_{\text{ctrl}} > 0, \quad V_{\text{nuis}} > 0, \quad V_{\text{cross}} \neq 0 \implies \text{Var}[\Delta s] = g^\top(V_{\text{ctrl}} + V_{\text{nuis}} + 2V_{\text{cross}})g. \tag{26}$$

**6. CPO case.** Since we fixed images and only varied conditions, we have

$$V_{\text{nuis}} = 0, \quad V_{\text{cross}} = 0, \quad V_{\text{ctrl}} = \text{Cov}(\widetilde{\Delta}_{\text{ctrl}}) > 0 \implies \text{Var}[\Delta s] = g^\top V_{\text{ctrl}} g > 0. \tag{27}$$

Note that we have [53]:

$$A \succeq B \implies g^T A g \geq g^T B g \tag{28}$$

Since covariance is always positively semi-definite, by comparing Eqs. 26 and 27, we can find that

the variance of CPO case is smaller since it only contains variance from perturbing the condition. Thus, the training objective only focuses on controllability.

Table 8: The optimal storage cost of one preference pair in terms of the number of RGB images (i.e. 1.66 means storing a preference pair is equivalent to storing 1.66 RGB images). Note that segmentation maps can be stored as labels, and poses can be stored as keypoints.

|  | With original image | Without original image |
|---|---|---|
| DPO | 3.66 | 2.66 |
| Ours | 1.66 | 1.66 |

Table 9: Training Hyperparameter.

| Prompter | Segmentation | | Pose | Canny | HED | Lineart | Depth |
|---|---|---|---|---|---|---|---|
|  | ADE-20K | COCO-Stuff | COCO-Pose | MultiGen-20M | | | |
| Learning rate | 1e-8 | 1e-8 | 1e-7 | 3e-9 | 1e-8 | 3e-9 | 1e-8 |
| Optimizer | AdamW [54] | | | | | | |
| $\lambda$ | 0.05 | 0.05 | 0.1 | 0.05 | 0.05 | 0.02 | 0.05 |
| $m$ | 0.01 | 0.005 | 0.01 | 0.005 | 0.01 | 0.05 | 0.01 |
| Number of steps | 10K | 10K | 20K | 10K | 10K | 10K | 10K |
| Batch size | 16 | 16 | 16 | 256 | 256 | 256 | 256 |

# D  Comparison to DPO dataset

To generate a DPO dataset, we need to sample $I^w, I^l$ using a generative model $G$ (e.g., ControlNet [7] or ConrolNet++ [11]). However, since both samples are drawn from the same distribution (i.e., the sampling process of $G$), their alignment with $\mathbf{c}$ may be nearly indistinguishable. In practice, we must sample multiple images and select the most and least aligned ones. In our experiments shown in Fig. 1, we sampled 20 images. We assume the controllability score $s$ of a generated image follows a distribution $\mathcal{S}$, where $\mathcal{S}$ denotes the distribution of controllability scores for all images sampled by model $G$. According to order statistics, for any distribution $\mathcal{S}$, if $s_1, \ldots, s_{20} \sim \mathcal{S}$, then $P\left(\min(\{s_n\}_{n=1}^{20}) < s' < \max(\{s_n\}_{n=1}^{20})\right) = \frac{n-1}{n+1}$. Here, $s'$ is a new sample drawn from $\mathcal{S}$. When $n = 20$, this probability is approximately $90\%$, implying that $I^w$ achieves a controllability score in the 90th percentile compared to a random sample $I$. Even under this setup, we still cannot observe clear differences in controllability from raw images, as shown in Fig. 1(c).

Moreover, in this case, DPO requires $20\times$ more computation than our method, which severely limits scalability. In practical applications, the storage requirement for the DPO dataset is also larger, as shown in Tab. 8. If the DPO dataset does not store the original image, our method requires only $\frac{5}{8}$ of the storage. If the DPO dataset includes original images to support regularization terms, our method reduces storage to $\frac{5}{11}$ of that required by DPO.

Beyond efficiency, our method may also mitigate safety concerns by storing only $\mathbf{c}^l$. DPO dataset curation involves generating full images, which can raise safety issues. In contrast, since our method involves only the generation of control signals (e.g., edges, depth, segmentation maps), it avoids such risks.

# E  Training Details

We include our training details on hyperparameters in Tab. 9. The training with batch size 16 and 10K steps takes approximately 8 hours to finish with 2 H100 GPUs. Readers may assume linear scaling when increasing the number of steps and batch size. Non-mentioned hyperparameters are in their default setup, such as optimizer configuration.

# F  More Ablation Studies

**Ablation Studies on DINOv2**. ControlAR [36] reveals the effectiveness of extracting image condition features with the DINOv2 model [37], which improves the image quality and controllability. We find that the DINOv2 is also effective in diffusion models, shown in Tab. 4 and Tab. 5. In general,

Table 10: More Ablation Studies on CFG Scales.

| CFG scale | Seg. (ADE-20K) | Seg. (COCO-Stuff) | Canny | HED | Lineart | Depth |
|---|---|---|---|---|---|---|
| | mIoU↑/FID↓/CLIP↑ | mIoU↑/FID↓/CLIP↑ | F1↑/FID↓/CLIP↑ | SSIM↑/FID↓/CLIP↑ | SSIM↑/FID↓/CLIP↑ | RMSE↓/FID↓/CLIP↑ |
| 1.5 | 45.18/28.28/30.95 | 34.21/17.16/31.05 | 39.49/20.45/29.54 | 0.8329/12.73/30.50 | **0.8607**/14.34/30.30 | **25.35**/17.54/29.65 |
| 3.0 | **46.40**/27.21/31.68 | 36.01/16.30/32.00 | **39.69**/18.95/30.75 | 0.8317/12.29/31.27 | 0.8584/13.28/31.13 | 25.67/14.71/31.34 |
| 4.0 | 46.38/27.59/31.80 | 36.10/17.01/32.20 | 39.68/**18.49**/31.15 | 0.8299/**12.24**/31.55 | 0.8559/12.98/31.44 | 25.98/14.62/31.72 |
| 7.5 | 44.81/30.30/31.97 | 35.49/19.30/**32.36** | 39.28/19.69/**31.83** | 0.8201/13.35/**32.07** | 0.8447/13.35/**31.98** | 27.49/15.88/32.31 |
| ControlAR (4.0) | 39.95/**27.15**/- | **37.49/14.51**/31.09 | 36.78/19.00/29.12 | 0.8184/14.03/30.82 | 0.7922/**12.41**/- | 29.33/17.70/29.19 |

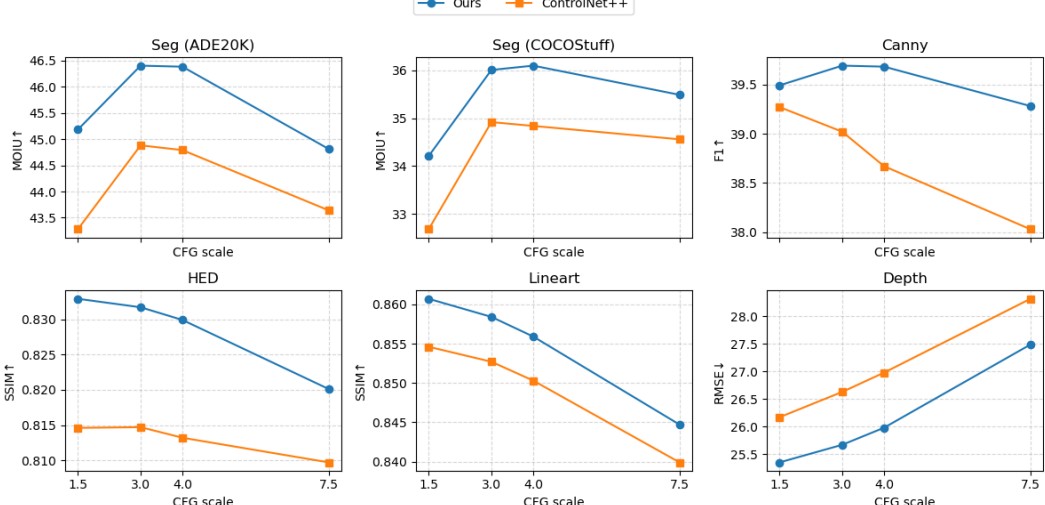

Figure 7: Comparisons with ControlNet++ under different CFG scales.

DINO-v2 feature extraction improves controllability, image quality, and text-to-image alignment in most scenarios.

**Additional Ablation Studies on CFG Scales.** We include additional ablation results in Tab. 10. We observe that after the CFG scale reaches 3–4, further increasing the CFG scale leads to a degradation in controllability. We also include a graphical illustration of controllability improvement under different CFG scales in Fig. 7. Under CFG scales 1.5, 3.0, 4.0, and 7.5, our method outperforms ControlNet++ with a notable margin.

**Visual Comparisons on Different Regularization Strengths.** As discussed, increasing controllability by adjusting the regularization strength leads to a slight drop in FID score, but the difference is not visually noticeable. Visual comparisons are presented in Fig. 8, where quality differences are minimal and difficult to perceive due to their subtlety.

# G  Additional Visual Examples

We include additional visual comparisons with SOTAs in Fig. 9 and visual comparisons between CPO and DPO in Fig. 11. We implement our method on FLUX-ControlNet for the Lineart task and include the visual comparisons between our methods and promeAI's open-sourced FLUX-ControlNet for Lineart in Fig. 10. The training data for FLUX-ControlNet is obtained from a subsample of LAION-5B [55]. To generate the losing condition, we use our CPO models to generate samples, detect conditions, and resize $c^l$ to 1024 pixels. For demo showcase, we include additional visual examples for Pose in Fig. 12, Segmentation in Fig. 13, HED in Fig. 14, Lineart in Fig. 15, Canny in Fig. 16, and Depth in Fig. 17.

**Failure Cases.** We present failure cases in Fig. 18. Our method struggles with extremely dense control conditions and with generating fine details for small human faces—a common limitation across all diffusion-based models.

**Gnerated Images & Extracted Condtions**

**Input & Condition**  $\lambda = 0$  $\lambda = 0.05$  $\lambda = 0.15$

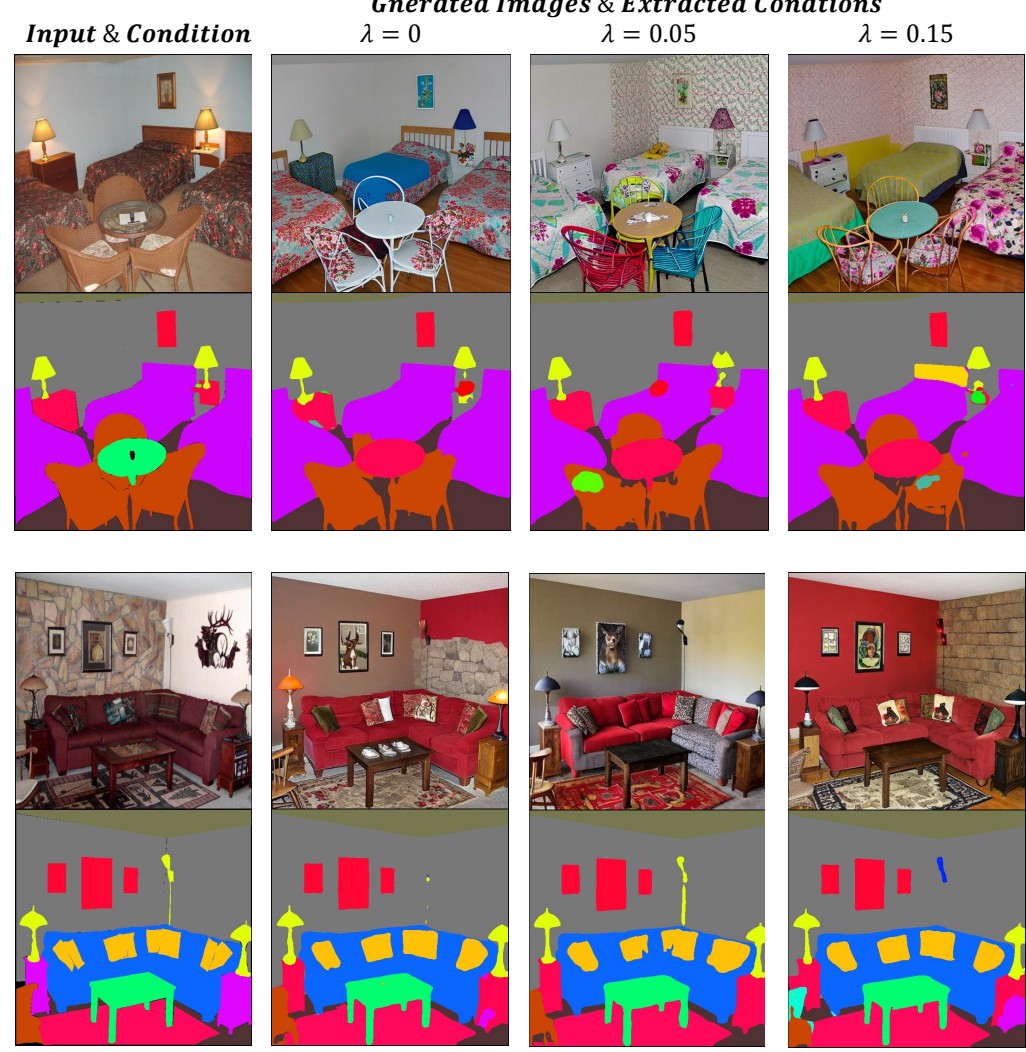

Figure 8: Visualization with different $\lambda$.

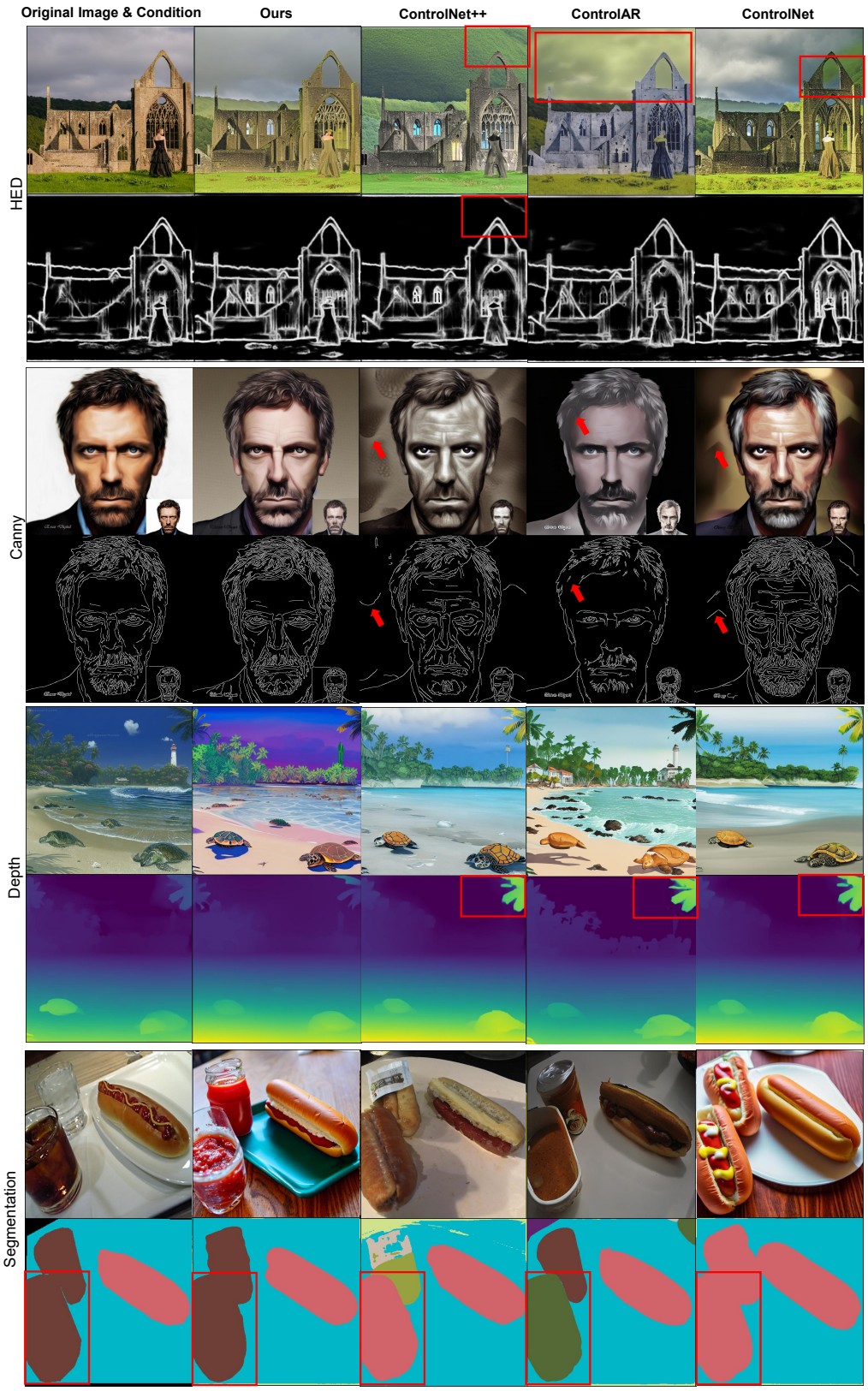

Figure 9: Additional Qualitative Comparison in Controllability. Red boxes indicate the area where our method achieves better controllability.

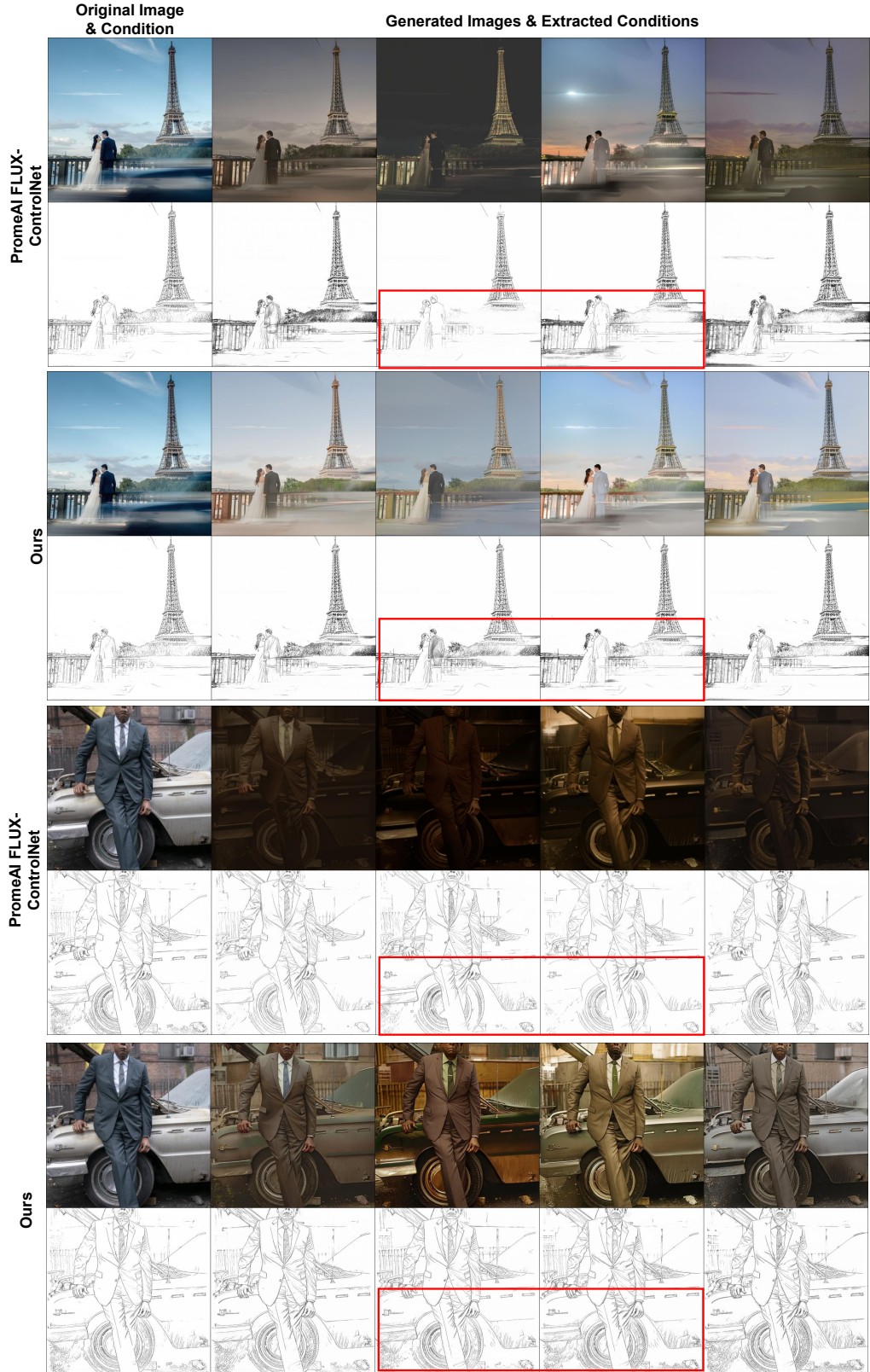

Figure 10: Comparison with PromeAI FLUX-ControlNet.

**Original Image**  **Pose**  **CPO**  **DPO**

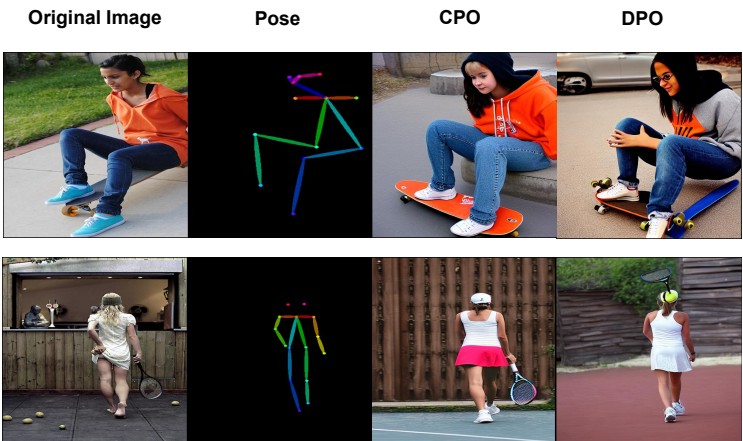

Figure 11: visual Comparisons between CPO and DPO in Pose task.

**Original Image**  **Pose**  **Generated Images**

Figure 12: Additional Visual Examples for Pose.

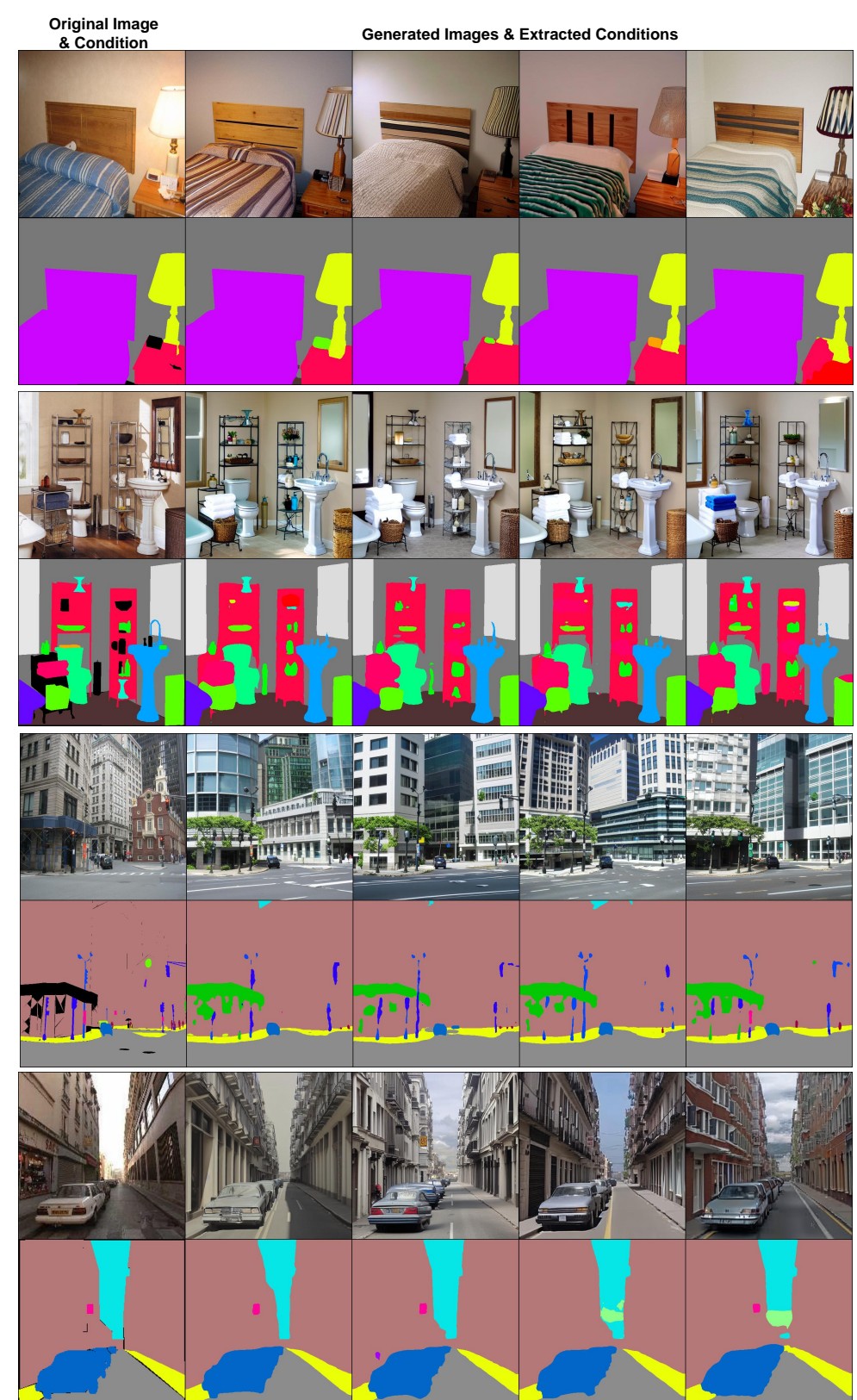

Figure 13: Additional Visual Examples for Segmentation maps. Note that black areas in ground truth segmentation are 'background' and not considered for correctness.

**Original Image & Condition**

**Generated Images & Extracted Conditions**

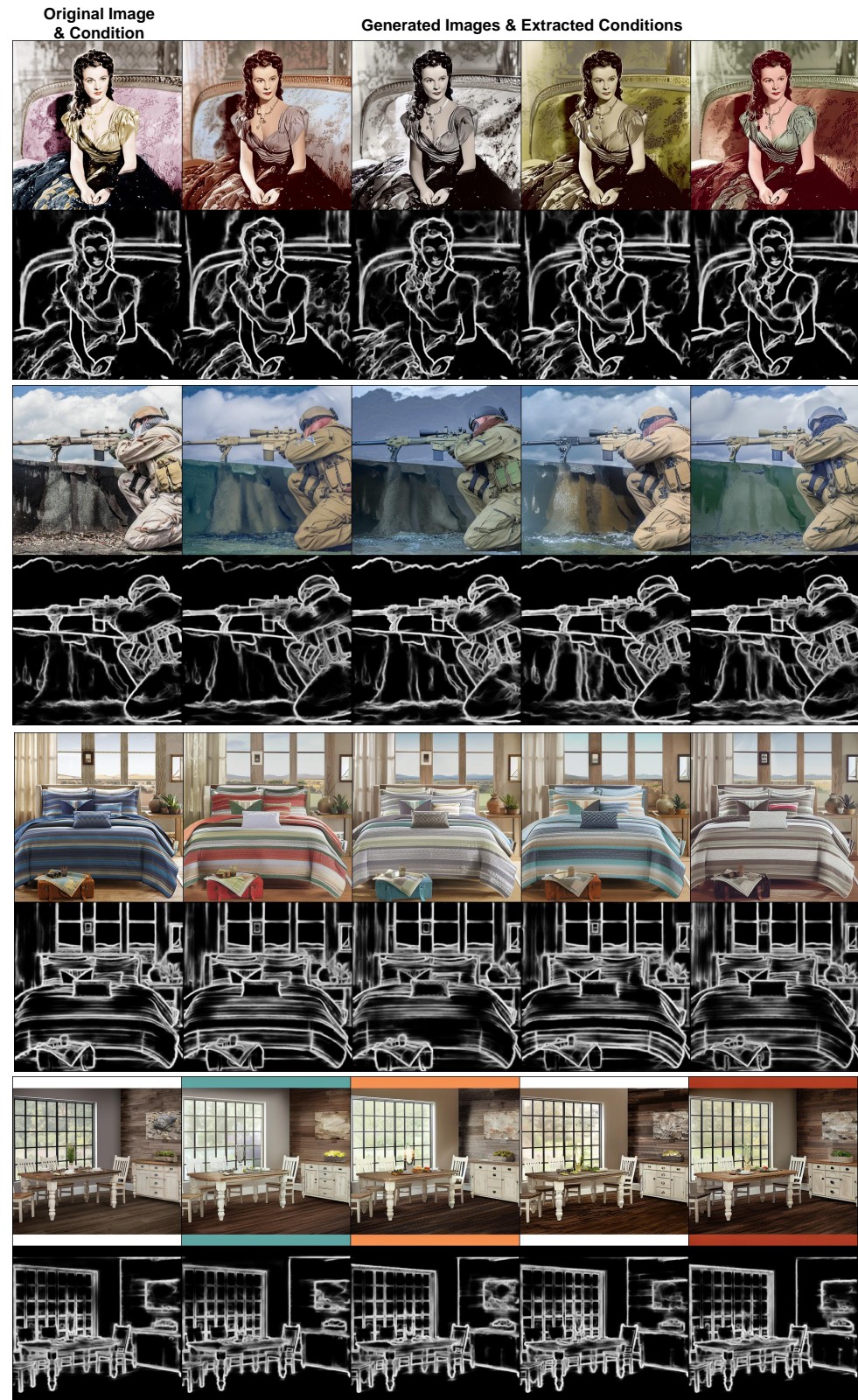

Figure 14: Additional Visual Examples for HED.

**Original Image
& Condition**

**Generated Images & Extracted Conditions**

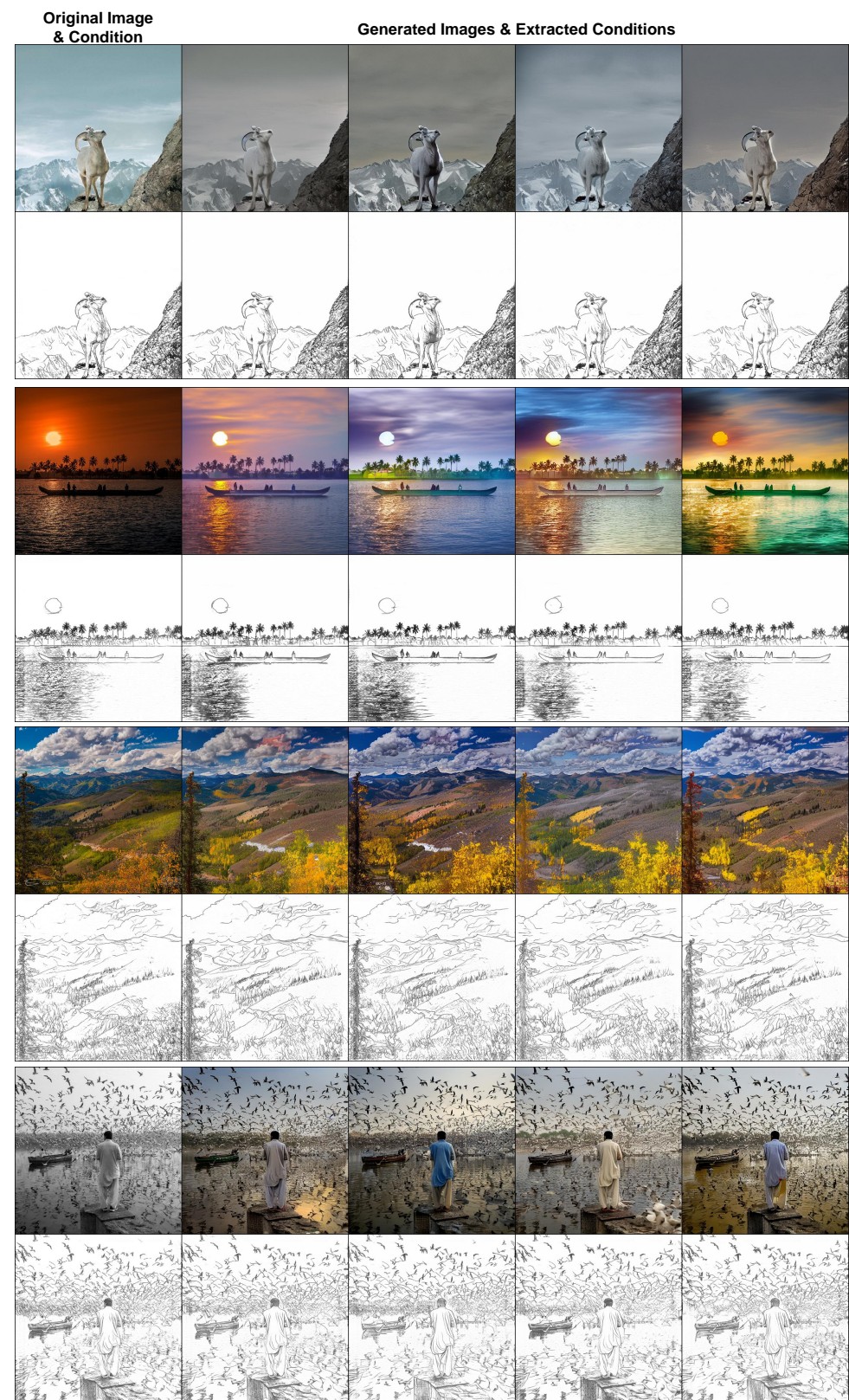

Figure 15: Additional Visual Examples for Lineart.

**Original Image & Condition**

**Generated Images & Extracted Conditions**

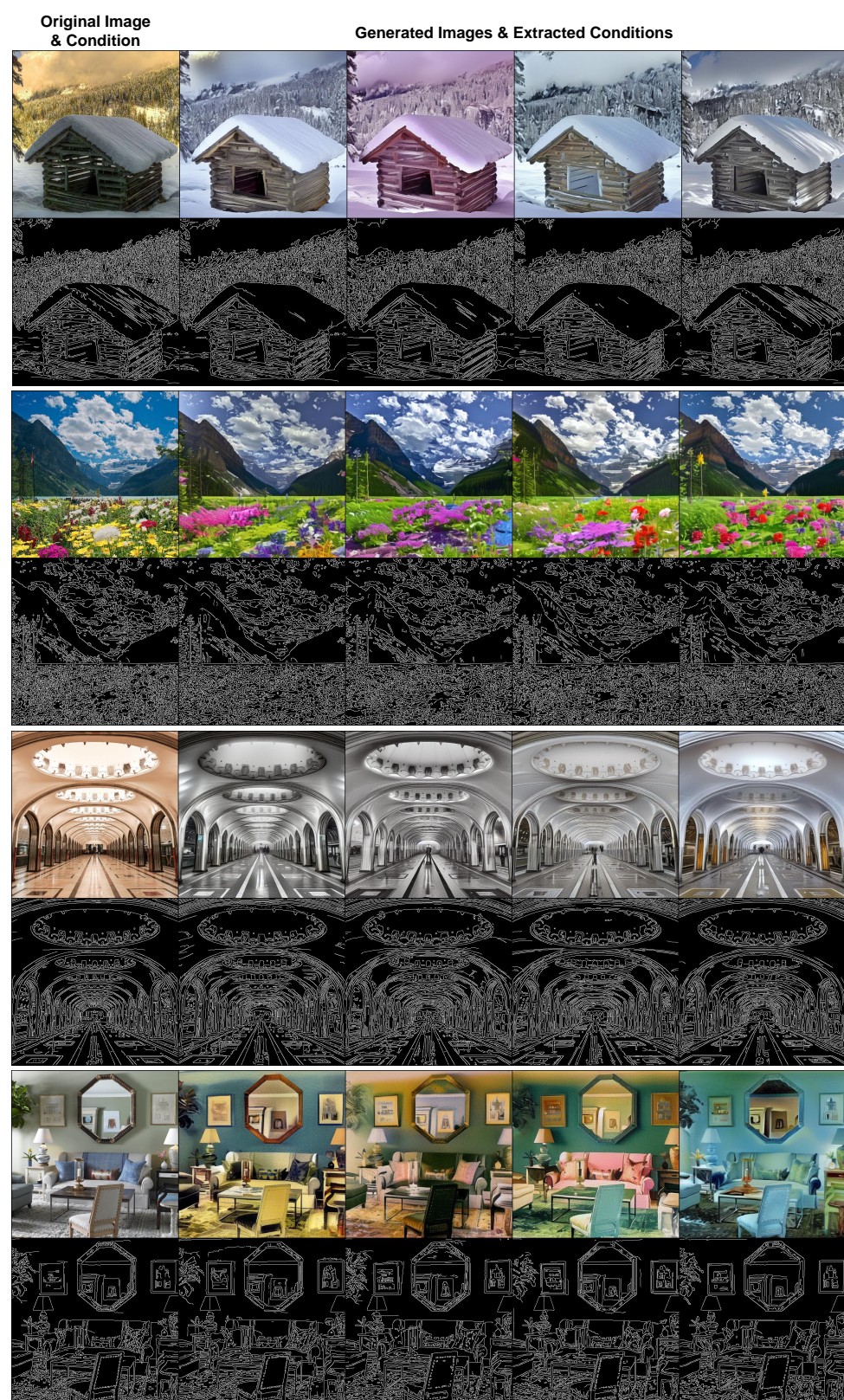

Figure 16: Additional Visual Examples for Canny.

**Original Image
& Condition**

**Generated Images & Extracted Conditions**

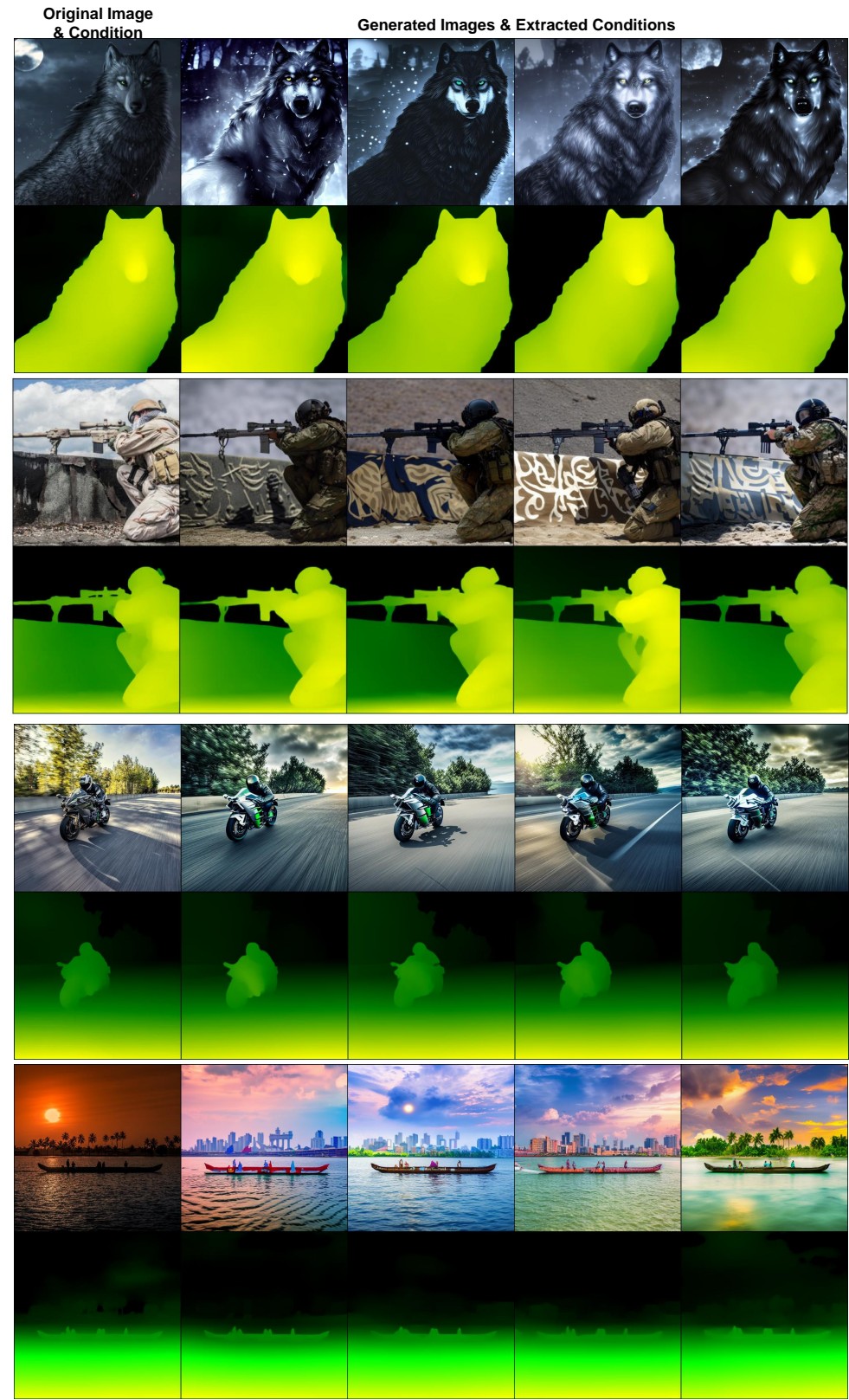

Figure 17: Additional Visual Examples for Depth.

**(a) Highly dense control signals**

Input and condition      Generated images and extracted conditions

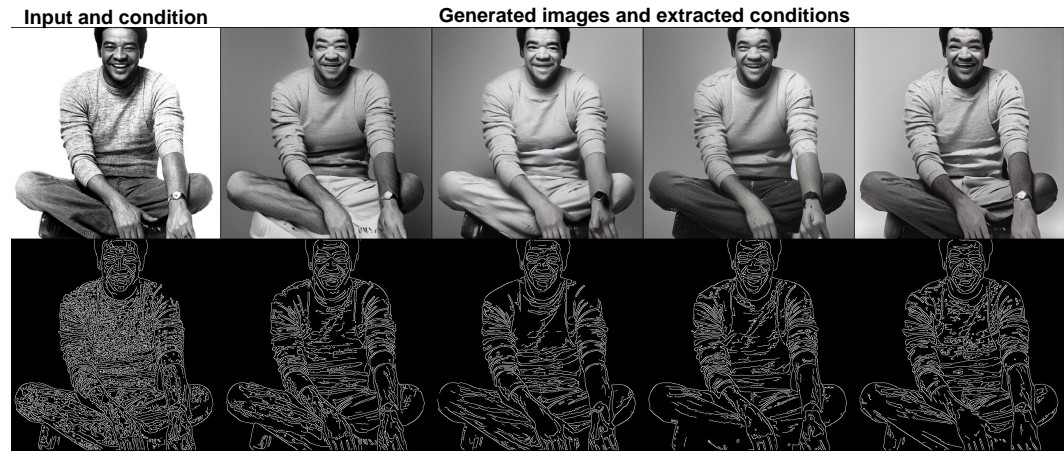

**(b) Small human faces**

Condition      Generated images

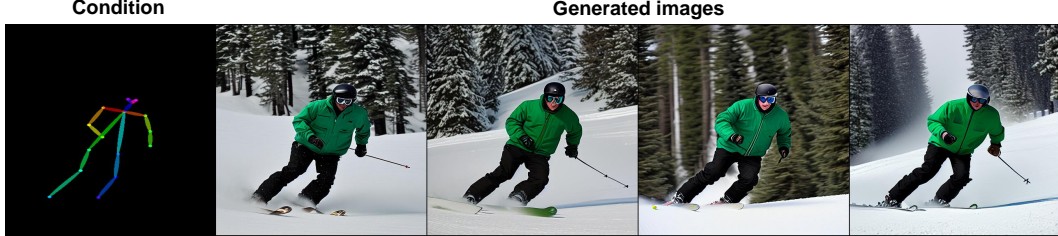

Figure 18: Failure cases. (a) When the condition is highly dense, the generated images can hardly follow it. (b) Similar to the problem with all diffusion models, our method struggles with human generation when humans are small but require details in their faces.

