# OpenReview forum: "CPO: Condition Preference Optimization for Controllable Image Generation"
_NeurIPS.cc/2025/Conference — NeurIPS 2025 poster_

### Official Review · Reviewer_YcYa · 2025-06-24

**Clarity:** 3
**Significance:** 3
**Originality:** 3
**Rating:** 4
**Confidence:** 4

**Summary:**

This paper proposes using reinforcement learning for ControlNet-based image generation. Prior work such as ControlNet++ optimizes only low-noise timesteps, which can introduce optimization errors. While Direct Preference Optimization (DPO) can optimize across all timesteps, it lacks fine-grained control, making it difficult to improve controllability without affecting other generation factors. To address this, the authors introduce a novel Condition Preference Optimization approach that constructs winning and losing control signals and trains the model to prefer the winning ones. Experimental results demonstrate superior performance over both DPO and ControlNet++ across various control types and evaluation metrics.

**Questions:**

While the method shows good controllability scores, it sometimes yields lower FID and CLIP scores. It would be helpful if the authors could discuss this trade-off and provide insights into how controllability affects image quality and alignment with the text.

**Ethical Concerns:**

["NO or VERY MINOR ethics concerns only"]

**Final Justification:**

My questions have been well addressed. I would appreciate a more detailed quantitative comparison between CPO and DPO, as well as additional visual comparisons. While another reviewer noted that the limited performance improvement might be a concern, I believe the proposed method shows noticeable improvements, as demonstrated in the tables and figures.

**Limitations:**

Yes.

**Quality:**

3

**Strengths And Weaknesses:**

Strengths
1. The proposed idea is novel and well-motivated. To the best of my knowledge, no prior work has attempted to apply the DPO method for conditional image generation following ControlNet. The authors also identify the challenges in directly applying DPO and introduce a new approach—Conditional Preference Optimization (CPO)—to address these issues.

2. The paper is generally well-written and easy to follow.

3. The proposed method achieves strong controllability scores, demonstrating the effectiveness of CPO. Additionally, the authors conduct thorough ablation studies comparing CPO with DPO and analyzing hyperparameter choices, which further strengthen the experimental results.


Weaknesses
1. Despite the novelty of the approach, it raises questions regarding the theoretical equivalence between the proposed CPO and the original DPO. As stated in the introduction, while the relationship between the control and the generated images may be bidirectional, directly switching to CPO changes the data distribution compared to DPO. If CPO is not strictly equivalent, some clarification or refinement in the introduction would be helpful to better explain the motivation and implications.

2. In practice, when applying perturbations, how can we ensure that the perturbed control signals extracted from generated images share a similar distribution with real control signals? The paper mentions that the perturbations are derived from the generated images—but are they of comparable quality to the original control signals? Why not directly perturb the original control inputs instead?

3. The authors should consider conducting a user study to better evaluate the results. Human judgment can provide a more reliable assessment of how well the generated images align with the control inputs, compared to simply comparing the extracted control signals from generated images. Based on the current results, the quality of the extracted control signals is highly dependent on the extraction method used. For instance, the depth map for ControlAR in Figure 4 does not appear meaningful. Additionally, the HED results represent particularly challenging cases, and I would not conclude that ControlAR performs worse than the proposed method in those examples. Furthermore, in Figure 5(a), the alignment between the human subject and the control input is not very accurate.

---

> ### Author Rebuttal · Authors · 2025-07-30
>
> Before our response, we would like to explain a little about our intuition so that the reviewer can have a clearer understanding. Simply speaking, we construct a preference pair $c^w$ and $c^l$ and let the model learn that $c^w$ aligns better to the input image than $c^l$, where the input image is from the real-world dataset.
>
> **Question:**
> Despite the novelty of the approach, it raises questions regarding the theoretical equivalence between the proposed CPO and the original DPO. As stated in the introduction, while the relationship between the control and the generated images may be bidirectional, directly switching to CPO changes the data distribution compared to DPO. If CPO is not strictly equivalent, some clarification or refinement in the introduction would be helpful to better explain the motivation and implications.
>
> **Answer:**
> 1. DPO and CPO are similar. They cannot be strictly equivalent due to their difference in formulation and intuition. They are similar in training loss formulation. In plain words, DPO does not want to sample worse images (i.e., keep the sampling trajectory far from the trajectory of worse images). This raises concerns mentioned in our intro, such as the inability to discern the condition from the image to perform meaningful preference. CPO deals with the condition instead. It tries to keep the sampling trajectory far from the less controllable trajectory. Therefore, it functions similarly to DPO. However, they are different in the way the CPO deals with the condition part (C) of a probability distribution $P(X|C)$, but DPO deals with the variable part (X).
> 2. We did not mention the data distribution change in CPO. We did mention that the distribution mismatch in the DDPM forward and sampling process causes some limitations for ControlNet++.
> 3. If the reviewer has concerns about the generation data distribution, then our method aligns better to real-world images than DPO. DPO requires high-quality model-generated data for training, and therefore, its training distribution diverges from the real-world image. However, we still keep the real-world images, but we contrast conditions. This may raise a concern about conditions distribution (the reviewer’s next question). But the condition distribution still aligns. For Canny, Lineart, Depth, and HED, conditions are extracted with the same mode for both winning and losing conditions, and therefore they are in the same distribution. For pose and segmentation, conditions are drawn with a pre-defined visualization method. For example, in segmentation, we map labels (argmax of output distribution) to predefined colors. In pose, we draw lines with predefined colors between keypoints. This process can also be understood as coloring models that map labels/keypoints, and the coloring models are identical for the winning (ground truth) condition and the losing condition. Therefore, the winning and losing conditions are in the same distribution because they are mapped to the image (pixel) space by the same model. Importantly, in the same distribution does not mean that the losing conditions extracted from generated images are correct. They will be incorrect due to the non-perfect controllability, which is what we desire to construct the losing condition $c^l$.
>
> ---
>
> **Question:**
> In practice, when applying perturbations, how can we ensure that the perturbed control signals extracted from generated images share a similar distribution with real control signals? The paper mentions that the perturbations are derived from the generated images—but are they of comparable quality to the original control signals? Why not directly perturb the original control inputs instead?
>
> **Answer:**
> We include the response for the distribution in our previous answer. For the quality of the control signal, please see our previous response. We intentionally make the extracted control signals from the generated images worse to perform contrastive learning so that the model knows the winning signal is better than the losing one. For perturbing the original control inputs, we did think of doing such a processing at the beginning. However, the issue is that we need to carefully consider the richness of perturbation. Here are some problems that we encountered and found too hard to solve. Take the pose as an example, the direction and distance that we should move each keypoint is hard to determine. The perturbed keypoints look reasonable, so that the model can learn meaningfully from telling winning conditions are better? For edges, how can we perturb edges? Direct shifting to left/right does not change the shape of edges.  Given these challenges, we came up with our method. The perturbation process will be automatic, and the perturbed conditions still seem reasonable.
>
> ---
>
> **Question:**
> The authors should consider conducting a user study to better evaluate the results. Human judgment can provide a more reliable assessment of how well the generated images align with the control inputs, compared to simply comparing the extracted control signals from generated images.
>
> **Answer:**
> We conduct a user study on hed and depth with controllability. We randomly select 25 groups of images for each task and ask 15 annotators to annotate.
>
> Here are the results in win-rate(%):
>
> | Task| CPO  | ControlNet++| ControlAR|
> |----------|-----|------------|-----------|
> |HED |47.73% |   37.87%   | 14.40%  |
> |Depth |46.67% |  40.27%  | 13.06% |
>
> We see that we outperform ControlNet++ by 10 percentage points in HED and 6 percentage points in Depth.
>
> ---
>
> **Question:**
> While the method shows good controllability scores, it sometimes yields lower FID and CLIP scores. It would be helpful if the authors could discuss this trade-off and provide insights into how controllability affects image quality and alignment with the text.
>
> **Answers:**
> The only time the FID and CLIP are apparently lower is on HumanArt. We mentioned it in lines 270-273 in our main paper. The drop in FID on this dataset is not due to lower quality but to a data distribution shift. Our model was fine-tuned on the realistic COCO-Pose dataset, whereas HumanArt contains numerous cartoons and artistic paintings. Fine-tuning shifted our model's output distribution closer to real-world photos, naturally increasing the FID distance to the artistic style of HumanArt. This demonstrates that our model successfully learned the target data distribution. A similar pattern was observed in GLIGEN's results, as it was also trained on COCO-Pose only. CLIP score is impacted in a similar way. We will add examples to showcase it for a better explanation in our final version.
>
> For any other datasets, the changes in FID/CLIP are ignorable due to the inherent randomness of generation models. If you evaluate the dataset twice with different environments, machines, random seeds, etc, results will be slightly different. Therefore, it is inconclusive for the FID/CLIP on other datasets.
>
> However, we do observe that, if we do not use regularization, the controllability will increase further with worse FID and CLIP, and we do not visually find a strong difference in image quality (we kindly refer the reviewer to Figure 7 in our appendix). One reason is that the model tries to overfit on controllability, and some unobservable features might be captured by FID/CLIP, which is the reason we use original training to regularize.

---

> ### Author Response · Authors · 2025-08-04
>
> Dear Reviewer YcYa,
>
> We appreciate your constructive comments and are trying our best to improve our finalized version. To ensure we have successfully addressed your concern, we would like to know if the explanation on the difference between CPO and DPO sounds reasonable to you. If you have any other concerns, we are more than happy to address them promptly.
>
> Best, 17633 Authors

---

> > ### Comment · Reviewer_YcYa · 2025-08-06
> >
> > Thank you to the authors for the response. My questions have been addressed. I have also read the other reviews and responses.
> >
> > - Reviewer HcdE raised a question about using different pose detectors during data curation, and I appreciate that the authors provided evidence showing that the results remain similar. The reviewer also mentioned the comparison between CPO and DPO. Is there a quantitative evaluation for this?
> >
> > - The response to Reviewer yv4M regarding the difference in condition pairs makes sense to me, and I appreciate that the authors are conducting an additional experiment using a more advanced backbone.
> >
> > - Reviewer Q7TW’s question about the comparison involving longer training time and the combination of CPO and DPO also seems to be addressed. While their main concern might be the limited performance improvement, I find the gains to be noticeable both quantitatively and qualitatively.

---

> ### Author Response · Authors · 2025-08-06
>
> We appreciate the reviewer for the efforts and comments. The reviewer's concerns are addressed. After reading through other reviewers' comments, the reviewer also believes that our responses to other reviewers' comments address their concerns.
>
> The quantitative comparisons between CPO and DPO are shown in Figure 6. We plot the mAP and FID values under different training steps of CPO and DPO. Our method is plotted on the upper left side of DPO, showing superior performance. For example, when the FID value of the evaluated model is around 39.5, our method achieves mAP 87.98% but DPO gets 86.90%.
>
> We are still actively working on improving our final version and addressing existing concerns.
>
> Best,
>
> 17633 Authors

---

### Official Review · Reviewer_yv4M · 2025-07-01

**Clarity:** 3
**Significance:** 3
**Originality:** 3
**Rating:** 5
**Confidence:** 3

**Summary:**

This paper aims to enhance the controllability of image generation using ControlNet. It begins by examining the shortcomings of existing methods based on Direct Preference Optimization, which often struggle with uncertainties inherent in generative models—specifically, a winning image may compromise on other factors such as overall image quality. To address these challenges, this paper proposes Conditioned Preference Optimization, which constructs preference condition pairs rather than relying solely on image pairs. The proposed method demonstrates superior performance compared to previous approaches.

**Questions:**

For the condition pair, how to control the differences? If the difference is very large, it might be invalid. The authors should detail the process of constructing the pair, such as pose, canny, depth, segmentation ,etc.

The backbone is still SD1.5. To better validate its effectiveness, please use SDXL, even SD3 and FLUX.

**Ethical Concerns:**

["NO or VERY MINOR ethics concerns only"]

**Final Justification:**

Thank the authors for their responses. I agree that due to the time limitation, the additional experiments are hard to complete. As the authors have made efforts to give explanations, I keep my initial rating.

**Limitations:**

Yes

**Quality:**

3

**Strengths And Weaknesses:**

Strength:

The proposed method has clear motivation and analysis, providing analysis for previous methods.

The proposed method is simple but effective, with similar formulation to previous methods.

The paper also provides theoretical proof for the proposed method, lending it greater validity.

This paper is well-written. For example, the Fig.1 and Fig.2 can clear illustrate the motivation and strategy.

Weaknesses:

For the condition pair, how to control the differences? If the difference is very large, it might be invalid. The authors should detail the process of constructing the pair, such as pose, canny, depth, segmentation ,etc.

The backbone is still SD1.5. To better validate its effectiveness, please use SDXL, even SD3 and FLUX.

---

> ### Author Rebuttal · Authors · 2025-07-30
>
> **Question:**
> For the condition pair, how to control the differences? If the difference is very large, it might be invalid. The authors should detail the process of constructing the pair, such as pose, canny, depth, segmentation ,etc.
>
> **Answer:**
> We currently do not have a process to control the difference. The reviewer’s comments are helpful and may possibly provide a way to improve our work. Our pair construction process is described in lines 195-200.
>
> 1. We first obtain the ground truth (segmentation/pose) or oracle (canny, lineart, hed, depth) as $c^w$.
> 2. Then, we generate an image $I$ with ControlNet++ controlled by $c^w$.
> 3. Finally, we use the condition detector to extract $c^l$ from $I$.
>
> For example, in the canny edge task, the condition detector is just a canny edge detector.
>
> Since ControlNet++ achieves strong controllability, we assumed the difference would not be large. However, this is more about curating a better dataset, and we will follow this direction and try to improve our dataset by adding some difference-control mechanisms.
>
> ---
>
> **Question:**
> The backbone is still SD1.5. To better validate its effectiveness, please use SDXL, even SD3, and FLUX.
>
> **Answer:**
> We are actively working on FLUX training. However, since FLUX requires $1024 \times 1024$ resolution images as inputs, we collected a high-resolution dataset first. We filtered out 3M images from the LIAON-Aesthetic-5+ dataset with an aesthetic score higher than 5.6, then we captioned all images with qwen-2.5-vl-32b for a high-quality text description. We are currently training FLUX-ControlNet, but FLUX takes weeks to train with 64 A100 80G gpus, and we only finished the first 5K steps of training FLUX-ControlNet in lineart task, but we do not have results now. It will take one more week to finish FLUX-ControlNet, if everything works as expected. Then we can proceed to CPO. We will include available results in our final version.

---

> > ### Comment · Reviewer_yv4M · 2025-08-08
> > **Thanks for response**
> >
> > Thank the authors for their responses. I agree that due to the time limitation, the additional experiments are hard to complete. But, regarding Q1, I hope the authors could give more reasonable analysis.

---

> > > ### Author Response · Authors · 2025-08-08
> > >
> > > Thank the reviewer for the comments. We will include additional experiments that compare CPO with or without a difference control to see how the performance differs. For example, we can generate 5 images and extract 5 conditions. Then we pick the best one as the $c^l$ to control the difference. We are now working on the experiment on ADE_20K to see if there is any improvement. We will update as soon as possible.
> > >
> > > Best,
> > >
> > > 17633 Authors

---

> > > ### Author Response · Authors · 2025-08-08
> > > **Follow-up on comments**
> > >
> > > While we are running experiments, we would like to let the reviewer know that our training algorithm automatically filters out the *invalid* cases if the difference is too large.
> > >
> > > Based on Equation (10) in our main paper, the gradient direction of CPO is controlled by $max(d_{\theta} + m,0)$, where $d_{\theta}  = \|\epsilon - \epsilon_{\theta}(x_t,c^w,t)\|^2 - \|\epsilon - \epsilon_{\theta}(x_t,c^l,t)\|^2$. If $c^w$ is significantly better than $c^l$, we would observe that $\|\epsilon - \epsilon_{\theta}(x_t,c^w,t)\|^2 < \|\epsilon - \epsilon_{\theta}(x_t,c^l,t)\|^2 + m$ and therefore the gradient is zeroed out. We experimentally set $m$ to be a small number (usually 0.01 to 0.001).
> > >
> > > Though we intentionally bring this formulation to avoid overly contrasting $c^w$ and $c^l$ (when the model prefer $c^w$ over $c^l$ more than some margin, the gradient for this example is 0) because $c^l$ is not completely incorrect, this formulation also filters out the training examples where the difference is too large (the reviewer is concerned about).
> > >
> > > We will update experimental results to support our argument by further editing this response. We hope that the reviewer's question regarding the differences being too big could be resolved.
> > >
> > > Best,
> > >
> > > 17633 Authors

---

> ### Author Response · Authors · 2025-08-04
>
> Dear Reviewer yv4M,
>
> We appreciate your constructive comments and are trying our best to improve our finalized version. To ensure we have successfully addressed your concern, we would like to know if the details provided address your concerns. We are still training the FLUX ControlNet for Lineart, which is currently 6000+ steps. We inspect the generation quality at the 5000th step visually and find it on the right track (i.e., quality is good). However, it still needs more steps to converge. We should be able to provide more results in the final version. If you have any other concerns, we are more than happy to address them promptly.
>
> Best,
> 17633 Authors

---

> ### Author Response · Authors · 2025-08-09
> **Follow up to yv4M**
>
> Dear Reviewer yv4M,
>
> We have finished the experiments on selecting the best of five $c^l$ on ADE20K. Our original method achieves MIOU 44.81. With the new dataset that we control the difference from being too large, the MIOU is 44.83. Since the generation results are subject to randomness, such a difference is inconclusive. Therefore, our algorithm successfully overcomes such a problem by itself (see previous response on Equation (10)) without needing to control the difference between $c^w$ and $c^l$. We thank the reviewer for comments and follow-ups, and are trying our best to improve our final version. We hope that additional experiments can address the reviewer's concern.

---

### Official Review · Reviewer_Q7TW · 2025-07-01

**Clarity:** 3
**Significance:** 2
**Originality:** 3
**Rating:** 4
**Confidence:** 4

**Summary:**

The paper introduces a new Condition Preference Optimization (CPO) method that aligns the conditions directly with DPO instead of using images for controllable image generation. It uses the original condition as the winning condition while treating the generated condition as the losing condition. Experiments are conducted on segmentation, depth, canny, edge and pose conditions across 5 different benchmarks.

**Questions:**

How does the performance vary with different detectors? How many images are sampled to obtain the losing condition? Is it just one?

How does the performance compare to training baseline models for longer (same amount of time as CPO) with the new data? Is the improvement solely because of CPO?

What is the effect of this alignment on the background regions? The model’s performance is specifically optimized to improve the controllability in the regions specified by the conditions. Does it degrade the performance for those regions outside the conditions?

Will there be any interference when CPO is performed on top of DPO or vice-versa?

**Ethical Concerns:**

["NO or VERY MINOR ethics concerns only"]

**Final Justification:**

I appreciate the authors in providing additional experimental evaluations that addressed most of my concerns on error propagation, performance improvement and training time. Overall, the method shows good empirical gains over the baselines but it still suffers from the limitations of the original DPO method such as the noisy preference issue. A more thorough discussion and analysis on the noisy preference problem in the paper will be beneficial. Therefore, I increase my rating to borderline accept (4).

**Limitations:**

Yes

**Quality:**

3

**Strengths And Weaknesses:**

**Strengths**: The paper proposes a novel method for improving the alignment and controllability in conditional image generation with the CPO method that uses conditions as preference images instead of their RGB counterparts. Experimental results show improved performance over the baseline ControlNet or ControlNet++ across pose, depth, segmentation and sketch conditions.

**Weaknesses**:
The proposed method uses a discriminative model (condition detector) to obtain the losing conditions from the generated images of the baseline model. However, this assumes that the detector is perfect which is not the case in practice. This results in error propagation to the CPO where even good images will be rejected. So, it is sensitive to the detector used. How does the performance vary with different detectors? How many images are sampled to obtain the losing condition? Is it just one?

The proposed method requires a strong pretrained model as initialization such as the ControlNet or ControlNet++ and so the computational complexity increases. How does the performance compare to training baseline models for longer (same amount of time as CPO) with the new data? Is the improvement solely because of CPO?

 The performance improvements when evaluated under the same CFG configuration of 7.5 listed in Tables 1 and 2 are not very significant compared to the additional complexity involved in training. What is the performance difference between the methods when using the CFG scale of 4.0? It is not clear why the performance changes a lot at CFG 7.5 especially since the initialization is from a base model that uses a CFG of 7.5.

What is the effect of this alignment on the background regions? The model’s performance is specifically optimized to improve the controllability in the regions specified by the conditions. Does it degrade the performance for those regions outside the conditions? This can also be seen from the FID scores reported in Table 2 where the performance is comparable or slightly inferior to the baseline models.

In practice, we require both controllability and high-quality images. So, preference along both dimensions is essential. Will there be any interference when CPO is performed on top of DPO or vice-versa?

---

> ### Author Rebuttal · Authors · 2025-07-30
>
> We thank the reviewer for the constructive comments. The reviewer does agree with our novelty.
>
> **Question 1: On detector sensitivity and error propagation.**
> The reviewer noted that our method uses a condition detector that is not perfect, which could lead to error propagation and the rejection of good images.
>
> **Answer:**
> Thank you for this question. Our method fundamentally avoids this issue for the following reasons:
> 1. We Contrast "Conditions," Not "Images": Our method (CPO) contrasts a "winning condition" with a "losing condition," not the images themselves. Both conditions are applied to the same input image. The winning condition is always the ground truth or an oracle. Therefore, the losing condition is, by definition, less aligned with the input image than the winning one. This mechanism prevents good conditions from being mislabeled as losers, thus eliminating the risk of error propagation.
> 2. Robustness to Different Detectors: To further validate this, we experimented with different detectors for segmentation (ADE20K), pose (COCOPose), and depth (MultiGen20M) tasks. As shown in the table below, the performance shows almost no difference, demonstrating our method's strong robustness to the choice of detector.
> 3. Yes, we only sample one image to obtain the losing condition.
>
> | Task & detctor    | Pose + YOLO11x| Pose + mmpose | ADE20K + UperNet-ResNet |ADE20K + UperNet-ConvNext |Depth + DPT-Large |Depth + MiDaS |
> |----------|------------|------------|------------|------------|------------|------------|
> | Our | 87.98 |   87.71 |  44.81   | 44.90 | 27.49 | 27.56|
> | Base Model| 72.53 |   72.53 |  43.64   | 43.64 | 28.32 | 28.32|
>
> ---
>
> **Question 2: Is the improvement solely due to longer training, not CPO itself?**
>
> **Answer:**
> The performance improvement stems entirely from CPO, not from longer training. We demonstrate this with the following experiments:
> 1. Extended Training of the Baseline: We trained the baseline model, ControlNet++, for the same training step as CPO. As shown below, it began to overfit on ADE20K (performance degraded) and showed negligible improvement on other tasks. In contrast, our CPO method consistently improves controllability without sacrificing FID.
> | Task    | ControlNet++| ControlNet++ more training | ours  |
> |----------|------------|------------|------------|
> | Seg (ADE20K)    | 43.64/30.24  |   40.91/32.57       |  44.81/30.30   |
> | Canny| 38.03/20.16 | 39.01/22.50  | 39.19/19.68    |
> | HED   | 80.97/15.01 | 80.81/14.92    | 82.01/13.35   |
> | Lineart  | 83.99/13.88  | 84.04/13.80    | 84.31/13.84    |
> | Depth | 28.32/16.66 | 28.15/16.78   | 27.49/16.64    |
>
> 2. Applying CPO on a Fully Converged Model: On the COCOStuff dataset, we train the baseline model (ControlNet++) until full convergence (5000 more steps) because we observe the under-convergence of ControlNet++. Further training beyond 5000 steps does not improve controllability without sacrificing FID a lot. The performance is approximately 90% of Oracle, which is extremely challenging to further improve. Even so, our CPO still achieved a noticeable improvement, proving its effectiveness beyond simple training extension.
> | Task    | Fully converged ControlNet++| ours  |
> | --- | --- | --- |
> | Seg(COCOStuff)     | 38.02/19.24 | 38.44/19.20 |
>
> ---
>
> **Question 3: Insignificant performance improvement and the effect of CFG-scale.**
>
> **Answer:**
> 1. On the Significance of Improvement: The improvement might seem modest for two main reasons:
>   - The baseline is already very strong: ControlNet++ achieves 80-84% of oracle performance on most tasks, leaving limited room for improvement.
>   - Balancing Controllability and FID: Our goal is to enhance controllability while maintaining or improving image quality (FID). As shown in the previous answer, longer training of ConrolNet++ either fails to maintain FID (canny) or improve further on controllability (other tasks), whereas CPO succeeds in achieving better controllability.
> 2. On CFG-scale: It is important to clarify that the CFG-scale is an inference-time technique, unrelated to model training or its parameters. It adjusts the strength of conditional guidance using the formula $pred = (c+1) \times pred_{cond} - c \times pred_{uncond}$. $pred_{cond}$ means the model output with full text input, and $pred_{uncond}$ means the model output with empty text input. Different CFG scales will thus naturally alter the output and the resulting evaluation scores. This is a standard practice for diffusion models and not a unique aspect of our method.
>
> **Question 4: Effect on background regions.**
>
> **Answer:**
> 1. Most Tasks Lack a Distinct "Background": In tasks like edge detection (Canny, HED, Lineart) and depth mapping, there is no strict "background" region. Every part of the image contains relevant conditional information. For example, the black region of edge detection tasks means **no edges** instead of **no requirement**.
>
> 2. Background Quality is Not Degraded: For tasks with backgrounds, such as segmentation and pose estimation, Figures 8 and 9 in our appendix clearly show that the quality of background regions is not degraded.
> 3. On the FID Score for HumanArt: The drop in FID on this dataset is not due to lower quality but to a data distribution shift. Our model was fine-tuned on the realistic COCO-Pose dataset, whereas HumanArt contains numerous cartoons and artistic paintings. Fine-tuning shifted our model's output distribution closer to real-world photos, naturally increasing the FID distance to the artistic style of HumanArt. This demonstrates that our model successfully learned the target data distribution. A similar pattern was observed in GLIGEN's results, as it was also trained on COCO-Pose only.  We will add examples to showcase it for a better explanation in our final version.
>
> ---
>
> **Question 5: Potential interference when combining CPO and DPO.**
>
> **Answer:**
> This is an interesting direction, but combining CPO with DPO presents practical challenges.
> 1. Challenges in Data Curation: CPO leverages the model's inherent imperfection in controllability, requiring only one generated image to obtain an effective "losing condition." In contrast, to ensure a "winning image" is superior in both quality and controllability, DPO would require generating and filtering a large number of samples, which is computationally prohibitive. Importantly, **DPO cannot be performed in quality only** since the alignment to control signals cannot be identical. However, CPO is performed for controllability only.
>
> 2. Negative Interference: Our experiments confirm this. We split our training steps equally for CPO and DPO. As shown in the table below, combining DPO and CPO, regardless of the order, results in performance inferior to using CPO alone, especially in terms of FID. The reason is that we apply a worse method on top of a better one. Figure 6 in our main paper shows that DPO performs worse than our method.
>
> | DPO   | CPO| CPO then DPO| DPO then CPO |
> |----------|-----|------------|------------|
> |87.64/41.28  |   87.98/39.94    | 87.89/41.07  | 87.94/41.00  |
>
> We do think this is an interesting idea for future improvement and research, and appreciate reviewers’ comments.

---

> > ### Comment · Reviewer_Q7TW · 2025-08-04
> >
> > Thanks to the authors for the rebuttal. It addressed some of my concerns (including combining CPO and DPO).  Following are my comments.
> >
> > 1. My point was that the relative ranking of the conditions are affected by the errors in the detector module especially because the method falls back to using the detector for groundtruth (winning) condition when the control signal is not available. A bad image might result in a good condition or vice-versa.
> >
> > 2. The performance improvement is a concern because CPO is expensive (training time, computations) while only offering limited improvements over prior ControlNet based methods.
> > My point on CFG scale was that different methods might have different optimal values for CFG and so it is essential to evaluate them on a range of values similar to the proposed method. For example, ControlNet++ might also provide better results for a different CFG scale.

---

> ### Author Response · Authors · 2025-08-04
>
> Dear Reviewer Q7TW,
>
> We appreciate your constructive comments and are trying our best to improve our finalized version. To ensure we have successfully addressed your concern, we would like to know if the experiments on DPO followed by CPO (or vice versa) were conducted reasonably to address your concerns. If you have any other concerns, we are more than happy to address them promptly.
>
> Best,
> 17633 Authors

---

> ### Author Response · Authors · 2025-08-06
>
> We thank the reviewer for the additional comments. Here are additional responses to the reviewer's comments.
>
> **A bad image might result in a good condition or vice-versa.**
>
> Please correct if we understand incorrectly. The reviewer may mean that the real ground truth condition $c^{gt}$ is not available. The method that we employ (detecting conditions from the ground truth image as the oracle $c^w$) to get $c^w$ and $c^l$ may result in a problem where $c^l$ is closer to the real ground truth than $c^w$. The reviewer points to *an existing concern in DPO: noisy preference*. Simply speaking, there may be some low-quality images that are labeled to be better than high-quality ones in the DPO dataset, and it is reasonable to have similar concern in CPO. We are more than willing to improve our method following the concern of noisy preference. *However, we would like to let the reviewer know that this is not a unique concern for our method but also for DPO.* Our goal is not to address such a problem (*this is a different research problem*). There are existing approaches that we can apply on top of our method to see if there is any improvement [1,2]. If we find these methods are effective, we will update them in our final version. However, we would like to claim that even if there might be a noisy preference problem, our method is still effective:
>
> First, our quantitative results (Table 1 in the main paper) and qualitative comparisons (Figures 4 and 5 in the main paper) show that it is effective.
>
> Second, our user study shows our method outperforms ControlNet++ in a subjective assessment.  We include a user study in HED and Depth in the following table. Therefore, even if there is such a problem, our method still outperforms our base model.
>
>
> | Task| CPO  | ControlNet++| ControlAR|
> |----------|-----|------------|-----------|
> |HED |47.73% |   37.87%  | 14.40%  |
> |Depth |46.67% |  40.27%  | 13.06% |
>
> **The performance improvement is a concern because CPO is expensive.**
>
> We kindly refer the reviewer to our previous response. When ControlNet++ cannot be further improved by increasing training steps, our method can further improve it. Moreover, when ControlNet++ achieves 80-90% of oracle results, any further improvement is challenging, but our improvement is still consistent.
>
> **ControlNet++ might also provide better results for a different CFG scale.**
>
> We include the evaluation results under the CFG scale 4.0 here. Under CFG scale 4.0, our method still outperforms ControlNet++. We will include more thorough evaluations with the CFG ranges, similar to our ablation studies, in our final version and visualize graphically. Depth metric (RMSE) is the lower the better, and remaining tasks are the higher the better
>
> | Task| ControlNet++ | CPO|
> |----------|-----|------------|
> |Seg (ADE20K) |44.79 |   46.38 |
> |Seg (COCOStuff) |34.84 |   36.10 |
> |Canny |38.67 |  39.68 |
> |Lineart |0.8503 |  0.8538  |
> |HED |0.8132|  0.8299 |
> |Depth |26.98 |  25.93  |
>
> We are actively evaluating with CFG scales 3 and 1.5 for ControlNet++ and will update accordingly. Since applying techniques regarding noisy preferences requires re-training, we will try to include them in the final version if they are effective.
>
> If the reviewer still has unresolved concerns, we are more than willing to address them.
>
>
> [1] Chowdhury, S. R., Kini, A., & Natarajan, N. (2024). Provably robust dpo: Aligning language models with noisy feedback. arXiv preprint arXiv:2403.00409.
>
> [2] Zhang, L., Liu, C., Xu, C., Hu, K., Luo, D., Wang, C., ... & Yao, Y. (2025). When Preferences Diverge: Aligning Diffusion Models with Minority-Aware Adaptive DPO. arXiv preprint arXiv:2503.16921.

---

> ### Author Response · Authors · 2025-08-09
> **Follow up to Q7TW**
>
> Dear Reviewer Q7TW,
>
> We have finished the evaluation of ControlNet++ on CFG scale 3 and 1.5, shown in the tables below. We can find that under all CFG scales, our method outperforms ControlNet++.
>
> CFG = 3
>
> | Task| ControlNet++ | CPO|
> |----------|-----|------------|
> |Seg (ADE20K) |44.88 |   46.40 |
> |Seg (COCOStuff) | 34.92 |   36.01 |
> |Canny |39.02|  39.69 |
> |Lineart |0.8527 |  0.8562  |
> |HED |0.8147| 0.8317 |
> |Depth |26.63 |  25.45  |
>
> CFG = 1.5
>
> | Task| ControlNet++ | CPO|
> |----------|-----|------------|
> |Seg (ADE20K) |43.28 |   45.18 |
> |Seg (COCOStuff) | 32.68 |  34.21 |
> |Canny |39.27|  39.49 |
> |Lineart | 0.8546 |  0.8580  |
> |HED |0.8146| 0.8329 |
> |Depth | 26.17 |  24.92  |
>
>
> Moreover, we also apply the Robust DPO method (an existing effective solution for the noisy preference problem, which is the reviewer's concern about bad images being labeled as good) on the depth task to deal with the noisy preferences issue. The reason we choose a depth task is that this is the task where noisy preference can mostly occur. The reason is that there is a ground truth depth by definition, but we cannot obtain the depth. For edges, there is no well-defined ground truth.
>
> Our method achieves RMSE 27.49, and after we apply Robust DPO to deal with noisy preferences, the result becomes 27.70. This implies that there might not be a noisy preference problem because the solution does not improve the performance.
>
> We thank the reviewer for comments and follow-ups, and are trying our best to improve our final version. We hope that additional experiments can address the reviewer's concern.

---

> > ### Comment · Reviewer_Q7TW · 2025-08-09
> >
> > I appreciate the authors in providing additional experimental results. Most of my concerns have been addressed. I will update my rating accordingly.

---

### Official Review · Reviewer_HcdE · 2025-07-03

**Clarity:** 4
**Significance:** 3
**Originality:** 3
**Rating:** 5
**Confidence:** 3

**Summary:**

The paper proposes the CPO (Condition Preference Optimization) method, which can improve the compliance of controllable image generation with control signals. The CPO method trains the model to prefer winning control signals by generating both winning and losing control signals. Compared to DPO, it requires significantly fewer computation for training data generation, making it more efficient. Compared to SoTA methods, it achieves better controllability. Under control signals such as segmentation and pose, it demonstrates a notable reduction in errors.

**Questions:**

- Is the improvement related to the correlation between the detector model for data generation and evaluation? If a different detector model is used during evaluation, will the same improvement still be achieved?  Taking Pose condition as an example, the CPO training data was generated using YOLO-11x-Pose, and the mAP calculation also used YOLO-11x-Pose. Could it be possible that visually, the pose of a person in the image appears incorrect, but YOLO-11x-Pose considers the pose more consistent? For the Pose mAP evaluation in Table 1, could another pose estimation model be used for assessment?
- The comparison with DPO is not sufficiently thorough. Could you provide more qualitative results?
- Could you also provide more qualitative comparative results?

**Ethical Concerns:**

["NO or VERY MINOR ethics concerns only"]

**Final Justification:**

My main concern is about overfitting to detection model for data generation. The rebuttal has successfully addressed this concern.

It is not possible to access more qualitative results, but given the large numerical improvement and the managable risk of overfitting, I tend to believe the qualitative quality without seeing more images.

The method is simple, effective and widely applicable.

Considering these factors, I decide to raise the rating to accept.

**Limitations:**

The authors adequately addressed the limitations and potential negative societal impact of their work. "yes".

**Paper Formatting Concerns:**

No paper formatting concerns.

**Quality:**

3

**Strengths And Weaknesses:**

strength
- The paper demonstrates a clear improvement in controllability, such as achieving  10%+ error rate reduction in segmentation, 70–80% error rate reduction in human pose, approximately 2–5% reduction on edges and depths after fine-tuning. This error reduction is relative to the model before fine-tuning.
- The writing of the paper is good, and it is quite readable. There is theoretical proof that the variance of CPO is smaller than that of DPO.
- The motivation of the paper is clear. The method is significantly more efficient than DPO, and easier to generate training data.

weakness
- The number of images for qualitative evaluation is too limited. In Section G, the focus is mainly on the generation results of this method, lacking more qualitative comparisons with other methods. The evaluation results are primarily based on the quantitative outcomes of  algorithms, without comparative assessment through a user study.
- There is a risk that training is favorable for a specific detection model, rather than improving the image information. The model used to generate data is the same as the one used for evaluation, such as YOLO-11x-Pose for pose estimation.

---

> ### Author Rebuttal · Authors · 2025-07-30
>
> We thank the reviewer for providing constructive comments. The reviewer appreciates our motivation, writing, and results.
>
> **Questions:**
> The number of images for qualitative evaluation is too limited, without comparative assessment through a user study.
>
> **Answer:**
> We’d like to let the reviewer know that we follow the practice of ControlNet++ and ControlAR to provide a qualitative Comparison (one group of visual comparison for each task (in the main paper), and several visual examples without comparison (supplementary Figure 8 - 13)). We are, of course, willing to provide additional qualitative results. However, this year Neurips does not allow us to upload a PDF with figures, nor the links. We will include more in the final version.
>
> For the user study, we conducted a user study on controllability on hed and depth. We randomly select 25 groups of images for each task and ask 15 annotators to annotate.
>
> Here are the results in win rate (%):
>
> | Task| CPO  | ControlNet++| ControlAR|
> |----------|-----|------------|-----------|
> |HED |47.73% |   37.87%  | 14.40%  |
> |Depth |46.67% |  40.27%  | 13.06%|
>
> In hed, we outperform ControlNet++ by 10 percentage points and 6 percentage points in Depth.
>
> ---
>
> **Questions:**
> Is the improvement related to the correlation between the detector model for data generation and evaluation? If a different detector model is used during evaluation, will the same improvement still be achieved? Could it be possible that, visually, the pose of a person in the image appears incorrect, but YOLO-11x-Pose considers the pose more consistent? For the Pose mAP evaluation in Table 1, could another pose estimation model be used for assessment?
>
>
> **Answer:**
> We do understand the reviewer’s concern that the improvement might be because we overfit to the pose detector. We provide two types of experiments to resolve the concern. First, our experiments show that changing the detector during dataset curation (i.e., the detector to construct $c^w$ and $c^l$) almost does not change the results. See the table below
>
> | Task & detector    | Pose + YOLO11x| Pose + mmpose | ADE20K + UperNet-ResNet |ADE20K + UperNet-ConvNext |Depth + DPT-Large |Depth + MiDaS |
> |----------|------------|------------|------------|------------|------------|------------|
> | Our | 87.98 |   87.71 |  44.81   | 44.90 | 27.49 | 27.56|
> | Base Model| 72.53 |   72.53 |  43.64   | 43.64 | 28.32 | 28.32|
>
> The differences are primarily due to the randomness of the generative model, and we can conclude that there is almost no difference.
>
> We also include mAP results where the dataset is curated with YOLO11x-pose but evaluated with a different model, mmpose (HRNet), below.
>
>
> |Methods| Cocopose | HumanArt|
> |----------|------------|-----------|
> | CPO| 95.48 |   67.29 |
> | Base Model| 91.34 |   60.44 |
>
>
> We can see that with HRNet, CPO still outperforms base models. HRNet tends to result in better evaluation results than YOLO11x-Pose. We will change the evaluation model to HRNet and update the entire table in our final version to resolve the reviewer’s concern.
>
> ---
>
> **Question:**
> The comparison with DPO is not sufficiently thorough. Could you provide more qualitative results? Could you also provide more qualitative comparative results?
>
> **Answer:**
> This year, Neurips does not allow us to provide any PDF or links. Therefore, we are not able to provide during the rebuttal. However, our structure follows previous SOTA ControlNet++ and ControlAR to provide one set of qualitative comparisons (in the main paper) and more visualizations with our models only (in the appendix). We will, of course, provide more qualitative comparisons in our final version, including CPO vs DPO and ours vs others.

---

> ### Author Response · Authors · 2025-08-04
>
> Dear Reviewer HcdE,
>
> We appreciate your constructive comments and are trying our best to improve our finalized version. To ensure we have successfully addressed your concern, we would like to know if the user study and changing detection models were conducted reasonably to address your concerns. If you have any other concerns, we are more than happy to address them promptly.
>
> Best,
> 17633 Authors

---

### Note · Authors · 2025-08-13

We thank the reviewers for their efforts and constructive feedback, as well as for their recognition of our work:

- **Novelty**: Reviewers Q7TW and YcYa acknowledge that CPO is a novel method for controllable generation.
- **Motivation**: Reviewers HcdE, yv4M, and YcYa agree that our work is clearly and well-motivated.
- **Clarity**: Reviewers HcdE, yv4M, and YcYa find the paper well-written and easy to follow.
- **Theoretical Analysis**: Reviewers HcdE and yv4M recognize the soundness and validity of our theoretical analysis.
- **Effectiveness**: Reviewers HcdE, yv4M, and YcYa agree that our method effectively improves controllability.

During the rebuttal and discussion phases:

- The concerns raised by **Reviewers Q7TW and YcYa** were addressed.
- **Reviewer yv4M** posed *follow-up questions* regarding the analysis of the differences between $c^w$ and $c^l$. We provided additional experimental results to address this, and we hope these have fully resolved the reviewer’s concern.
- At the request of **Reviewer HcdE**, we conducted a user study and provided additional experimental results, which we believe have addressed the reviewer’s concern.

In accordance with the rebuttal policy, we were unable to provide the requested qualitative comparisons during the rebuttal process; however, these will be included in the final version. We will also update the quantitative results for the pose estimation task using MMPose (HRNet). The final version will further include the user study, evaluation results with FLUX, and the performance of ControlNet++ under different CFG scales.

---

### Decision · Program_Chairs · 2025-09-17

**Decision:**

Accept (poster)

**Comment:**

The submission proposes Condition Preference Optimization (CPO), a new approach for enhancing controllability in image generation. Reviewers agreed that the paper is clearly written, well-motivated, and supported by sound theoretical analysis. Empirical results show consistent improvements across segmentation, pose, depth, and edge conditions, with notable efficiency advantages over DPO. The authors further strengthened their case through ablation studies, detector-robustness checks, and a user study, which confirmed better alignment compared to strong baselines such as ControlNet++. Some concerns were raised about limited qualitative comparisons, reliance on detector accuracy, and the use of SD1.5 rather than newer backbones. Additional experiments during the rebuttal—including cross-detector validation, CFG-scale analysis, and initial steps toward FLUX training—addressed these points effectively. While the performance gains are sometimes modest given strong baselines, the method remains simple, effective, and broadly applicable. Overall, the reviewers reached consensus that the paper makes a solid contribution to controllable generation research and warrants acceptance.